# A Trimming Design Method Based on Bio-Inspired Design for System Innovation

**Peng Zhang** [1,2]**, Xindi Li** [1,2]**, Zifeng Nie** [1,2]**, Fei Yu** [1,2,*] **and Wei Liu** [2,3,*]

1    School of Mechanical Engineering, Hebei University of Technology, Tianjin 300401, China;
     zhangpeng@hebut.edu.cn (P.Z.); lxd1615566854@163.com (X.L.); nzf970625510@163.com (Z.N.)
2    National Engineering Research Center for Technological Innovation Method and Tool,
     Hebei University of Technology, Tianjin 300401, China
3    School of Economics and Management, Hebei University of Technology, Tianjin 300401, China
*    Correspondence: fyu@hebut.edu.cn (F.Y.); 2020901@hebut.edu.cn (W.L.)

**Abstract:** The application of design knowledge determines the innovativeness of a technical scheme obtained by trimming (a tool for problem analysis and solving in TRIZ). However, limitations in the knowledge, experience and expertise of designers constrain the range of design knowledge that they can apply, thus reducing the effectiveness of trimming. In this paper, biological strategies are introduced to the trimming process to compensate for limitations imposed by the insufficient professional knowledge of designers, thereby improving design innovation. Therefore, this paper proposes a new design method that combines the trimming method and bio-inspired design (BID). First, a trimming analysis of the target system is carried out. Taking the missing functions of the trimmed system as a potential breakthrough point, a keyword search mode based on "V(verb)O(object)P(property) + the effect/features of the associated function" is used to search for biological prototypes in the biological knowledge base. Second, a fuzzy comprehensive evaluation method is used to analyze the biological prototypes from three dimensions, namely, compatibility, completeness and feasibility, and the best-matching biological prototype is selected. Finally, the biological solution is transformed into an engineering design scheme through a resource derivation process based on structure–function–attribute analogies. The proposed method can expand the range of design solutions by adding biological strategies as a new resource to solve trimming problems. The feasibility and effectiveness of the method are verified by redesigning a steel tape armoring machine.

**Keywords:** trimming; BID; innovative design; function design; design method

## 1. Introduction

In today's fiercely competitive market, companies largely depend on innovation to succeed, and refined design is one of the most important routes to achieving product innovations. As a method for solving refined design problems, trimming, which originates from the Theory of Inventive Problem Solving (TRIZ), has become a widespread method to realize product innovations. In the process of trimming, selected components of the system are removed, and useful functions are reallocated to residual system components or super-system resources [1]. The objective of trimming is to use existing resources after removing harmful components while streamlining design schemes, resulting in improved system ideality. However, solutions to trimming problems strongly depend on the knowledge and experience of designers, as trimming usually transforms contradiction problems into "how to" formulations that require the innovative use of resources to rebuild trimmed systems, which is a key design issue and a frequent difficulty for designers [2,3].

Trimming is applied to solve different types of inventive problems, from detailed component pattern improvements to system-level redesigns [4]. An innovative process that involves trimming requires more knowledge, potentially cross-industry and interdisciplinary knowledge, for problem solving [5]. However, existing resources are rarely directly

used to solve trimming problems due to limitations in designers' knowledge, thus hindering the generation of high-quality design solutions. When using the trimming method to solve innovative problems, engineering designers mainly rely on their professional experience to search for and identify suitable resources. In most cases, they need to retrieve potential knowledge from outside their professional fields, which is a difficult task.

Aiming to establish a method for identifying potential available resources after trimming, this paper combines trimming with bio-inspired design (BID). The objective of this method is to provide guidance for designers by introducing biological strategies to the innovation process so that they can quickly identify potential available resources and reallocate functions in the trimmed system.

Bio-inspired design (BID) is widely regarded as a feasible method for obtaining product design ideas based on biological systems, and it has yielded innovations in different areas [6–9]. Although it is a knowledge domain that differs from engineering design, biology can play an important role in stimulating breakthrough innovations in engineering. In order to increase the applicability of biological solutions to engineering problems, they are often processed and represented using function-based models, such as Design Analogy to Nature Engine (DANE) [10] or multi-biological effects (MBE) [11], which reveal the features of their functional design. The results of some studies [12,13] suggest that the application of biological solutions to an engineering design can indeed improve its novelty. Thus, biological knowledge can be used as a long-term resource to solve trimming problems.

However, a drawback of BID is that a practical scenario is necessary to overcome interdisciplinary obstacles before biological strategies can be transformed into technical solutions. Trimming offers advantages in scenarios involving transformed design problems. For example, after trimming invalid components from the system, "performance" problems can be transformed into "functionality" problems [3–5]. Therefore, trimming seems to be a feasible method for creating suitable design scenarios that facilitate the introduction of biological information to solve inventive problems.

In summary, this paper proposes a new design method that combines the trimming process with BID to solve the shortcomings of existing methods. First, a function analysis and Cause–Effect Chain Analysis (CECA) are used to analyze the target system, determine the trimming objects and their priorities, and identify missing useful functions after trimming. Second, a search strategy is constructed by first abstracting the function into search keywords. The keywords are used to search for the function in a biological prototype knowledge base. If multiple biological prototypes are obtained, then a fuzzy comprehensive evaluation method is used to select the one that best matches the target function. Finally, a multi-level analogy strategy based on structure–function–attribute mapping is used to derive available resources that exist within and outside the system and transform the biological solution into an engineering design scheme.

The contributions of this study are as follows:

1. An innovative design method is proposed in which trimming and BID are combined to complement each other. This approach compensates for limitations in designers' knowledge in the trimming process, optimizes the application of existing heuristic knowledge to the trimming process and improves the effectiveness of trimming.
2. The combination of BID and trimming improves the application of biological solutions to engineering problems and enhances the practicability of leveraging biological knowledge as an innovation resource.

The rest of the paper is organized as follows. Section 2 briefly reviews previous studies on trimming, BID and the integration of inventive problem solving and BID. Section 3 describes the theoretical framework of the proposed approach and presents its design workflow, and Section 4 is a case study that illustrates how the design problem is solved with the proposed method. Section 5 concludes the paper.

## 2. Literature Review

### 2.1. Trimming

In most cases, engineers are inclined to apply "addition" or "substitution" strategies to formulate their solutions, whereas trimming applies the opposite mechanism and uses the concept of "subtraction" to arrive at innovation [13,14]. Trimming is a method from TRIZ that is used to reduce the number of system components without losing system functionality, which, in turn, can improve the ideality level of the system [15,16]. The ratio of perceived benefits/(costs + harms) is frequently used as an indicator to judge whether the system ideality is improved [17]. The core principle of trimming is that indispensable functions of trimmed components can be carried out by existing resources in the trimmed system, thereby improving the ideality level of the whole system.

Function analysis [17] is applied to a technological system with functional interactions among components. Trimming usually serves as an important pre-process measure to analyze the products to be innovated. The systematic modeling method [18] is the foundation of trimming, as it is possible to cover both the functional and systematic information of the products under development. Trimming includes two significant tasks: the first is constructing the systematic model of the product and defining the components to be trimmed; the second is rebuilding the required functions with the remaining or newly added components. In practice, it is recommended to rank the priorities of components for trimming before making decisions on how to improve the overall ideality of the design [4,19]. The trimming priority is determined by the context of the design problem and its specific requirements, so it may vary when design scenarios are changed [20,21]. The algorithms applied in previous studies for calculating trimming priorities involve harmful function analysis [20], cost analysis [21], function rank analysis [1] and cause–effect chain analysis (CECA) [22], all of which are relevant to functional or systematic components.

In the first stage of trimming, several strategies are often used to identify the most significant components of the existing technological product system. An appropriate trimming plan is developed, which is an efficient but practical way to define inventive problems that need to be solved. Efimov-Soini and Chechurin [21] built computer-aided innovation software by incorporating the analysis of useful functions into the trimming process. A quantitative measurement of the trimming priorities of components was formulated by Yu [22] according to their ideality values, which were derived from a combination of value engineering and CECA. The cause–effect contradiction chain analysis (CECCA) was proposed by Sheu and Tsai [23], which improved CECA by incorporating engineering parameters and making it possible to consider factors that may cause contradictions when applying the trimming plan. Sheu et al. proposed a mathematical technique for integrating component groups to assist users in identifying priorities for trimming [24]. All of these studies have aimed to optimize the effectiveness of trimming while innovating improved products.

The second stage of trimming requires rebuilding the carriers of useful functions with the remaining resources in the system and super-system. A key step in this phase is searching for and identifying useful resources for rebuilding the system while following certain guidelines. The required functions are reallocated based on the functional analysis of system components [25–30]. San defined three rules for reallocating the required functions based on the functional relations between system components [25]. Sheu and Tsai extended the previous guideline by adding three rules to formulate a six-rule manual for trimming, resulting in a more systematic rule-based process [23]. To rebuild the useful functions of the system, Efimov Soini and Elfvengren [29] combined the three trimming rules and variable functional models that reflect changes in system architecture and functions. All of the aforementioned methods emphasize the importance of designers' expertise and require thinking outside the box to solve trimming problems. Therefore, the trimmed system has specific functional requirements, and designers who need to rebuild the system must search for and identify resources based on their background knowledge.

Innovative tools from TRIZ, such as its invention principles, can reveal key features of resources required for rebuilding post-trimmed product systems. Designers frequently turn to knowledge databases for appropriate resources. The TRIZ-based trimming method has provided several workable knowledge-searching strategies and has extended the search range. However, due to the high abstractness of this process, finding well-matched resources for rebuilding systems is still challenging for many designers. Therefore, a mechanism that offers searchable knowledge-oriented resources for rebuilding systems after trimming can be of great assistance in generating innovation solutions of high quality. In other words, the quality of work for system rebuilding can be improved by access to searchable knowledge resources that match functional requirements. From this perspective, BID is predicted to support the trimming process by providing abundant biological prototypes that meet the functional needs of the trimmed system and inspiring designers to develop creative solutions.

### 2.2. BID

BID significantly extends the searchable range of solutions to inventive problems by incorporating the biosphere. The core of BID lies in the application of biological principles to develop solutions to engineering problems [31]. Designs inspired by biological prototypes help to improve the sustainability of products [32], generate creative ideas [33] and even build new schemes for patents [34]. After decades of development, BID has become a sophisticated procedure for incorporating biological strategies into engineering designs for the purpose of enhancing innovation. The framework of BID involves three key processes: biological information representation, mapping biological to technological functions and the design process.

Function modeling has attracted considerable interest for representing biological information [35,36] since this format is familiar to engineering designers. There are two specific types of function-based representations of biological information: textual methods and graphic methods. In textual methods, biological strategies are represented by structural terms that include the main functional characteristics of biological prototypes. Asknature (http://asknature.org/aof/browse (accessed on 17 September 2020)) is one of the most widely used textual representation methods, and it also provides links to research papers or encyclopedia entries that provide more detailed information. Another method applied by BID mainly depends on graphic symbols to represent the relations between structures, functions and behaviors in biological systems. Design by Analogy to Nature Engine (DANE) [10] and State-Action-Parts-Phenomenon-Input-oRgan-Effects (SAPPhIRE) [36] are typical methods of graphic representation. Compared with the textual type, graphic models represent biological systems as functional models that are similar to functional structures in engineering design; however, model construction requires additional effort with this approach.

The use of function modeling methods is an effective strategy for mapping biological prototypes to technological solutions to meet their functional requirements. Functional characteristics typically play a vital role in bridging between the biosphere and technology. In practice, these characteristics are usually represented by a standard terminology set, such as a function basis [37], which is most widely used in function-orientated knowledge searching. An engineering-to-biology thesaurus [38], a variant of the function basis, summarizes the functional characteristics of a biological system to facilitate the process of BID. Besides the method of standard terminology, several description templates are formulated by combining verbs, nouns and other modified elements to characterize functions in biological prototypes. In addition, every type of representation model applies its own functional template: for example, DANE uses a template involving verbs, function carriers, objects and auxiliaries, while the SAPPhIRE (State-Action-Parts-Phenomenon-Input-oRgan-Effects) model uses a template that stresses the similarity of causal relationships between artificial and biological systems from seven aspects. In general, there are two types of BID algorithms, namely, solution-driven and problem-driven processes [36], which differ in

their starting points. A solution-driven process begins with biological principles and then explores their potential uses in engineering design, while a problem-driven process first analyzes the requirements of the engineering design problem and subsequently searches for and identifies matching biological prototypes that may solve it.

BID plays a significant role in the implementation of biological principles for inspiring creative solutions to engineering problems. However, it lacks a suitable scenario for innovation. Therefore, BID is usually embedded in other design processes while tackling the creative design process.

### 2.3. Previous Attempts to Integrate Inventive Problem Solving and BID

TRIZ was proposed for solving inventive problems, and it is able to create various scenarios for creative design. Therefore, several studies have attempted to integrate TRIZ and BID.

Bio-TRIZ [39] combines BID and the TRIZ approach to resolving contradictions. Bio-TRIZ summarizes the principles used by existing BID outputs and standardizes them as invention principles [40] that are then sorted into a PRIZM matrix [41]. Bio-TRIZ has also revealed that biology tends to solve contradictions by applying strategies that modify structures and information without involving changes in energy, which is usually manipulated to solve technological contradictions. Bio-TRIZ has expanded the application of TRIZ by introducing new design axioms and new design algorithms that apply biological approaches to solving contradictions. However, the biological solutions suggested by Bio-TRIZ are usually principles, which are sometimes too abstract to apply to detailed design tasks in practice.

MBE [11] (multi-biological effects) is another method that integrates the idea of effects in TRIZ, biological coupling [42] and BID, and it is able to support design tasks that involve multiple functional requirements [43]. MBE was proposed for integrating biological strategies into engineering design, and it contains several methods and tools that inherit the characteristics of TRIZ, biological coupling and systematic product design theory for engineering designers [44]. However, owing to its close connection to TRIZ, MBE also requires its users to have sufficient knowledge and skills related to conceptual design and TRIZ.

Similarly, as a feature of biological systems, function sharing encourages engineers to develop creative engineering systems [6,45] and enables a single structure to perform multiple functions at the same time [46,47]. Bhasin et al. derived the principles of bio-inspired product architecture that enable engineers to identify function-sharing scenarios at an early stage of product design and reduce the need to imitate biological structures for function sharing [48]. Furthermore, abstracting and simulating the product architecture of a biological system with shared functions promotes the problem-driven biological inspiration of function sharing; thus, function sharing in a biological design can be effectively utilized in engineering design [49]. However, the above method requires designers to have sufficient biological and design knowledge for extensive abstract work, which is difficult. The related research on bio-inspired function sharing has a degree of similarity to the BID-based trimming method proposed in this paper, which provides a function-based scenario for BID and connects design problems to biological solutions through function-based analysis. However, the two approaches focus on slightly different aspects. The above research aims to facilitate the abstraction of bio-inspired function-sharing scenarios by introducing a simplified function-method tree. However, the BID-based trimming method proposed in this paper introduces biological knowledge to the trimming process in an aim to address difficulties in identifying potential resources after trimming due to limitations in designers' knowledge or experience.

### 2.4. Summary of Review

The aforementioned discussion reveals that trimming is a continuously developed design approach to innovation. However, its effectiveness partly depends on the knowledge

and experience of designers, which may have limitations [3–5]. Therefore, an approach that helps engineering designers use knowledge outside of their field is very important for improving the effectiveness of trimming. With biological representations and a corresponding database, BID can serve as an important source of super-system resources to solve trimming problems, especially when functions are reallocated after harmful components have been trimmed. Consequently, the existing trimming design approach should be adapted to provide a design scenario in which biological knowledge is accessible by engineering designers. The proposed design approach in this paper is an output of a combination of trimming and BID, which provides a function-oriented design scenario to apply biological knowledge to trimming problem solving.

## 3. Theoretical Method

With the purpose of exploring the benefits of the integration of trimming and BID, this paper presents a design method that embeds BID in the process of trimming. On the one hand, BID can provide abundant knowledge for redistributing the useful functions of trimmed components. On the other hand, trimming improves the application of BID for solving creative problems by setting clear functional requirements for biological solutions. The proposed method is a novel contribution to this research domain, as it emphasizes the use of biological strategies to obtain solutions of high quality. Moreover, this approach also enriches the traditional BID method by incorporating trimming and provides applicable scenarios for the utilization of biological solutions. Finally, methods and tools from both BID and trimming can provide the assistance necessary to build the framework of this new method.

As shown in Figure 1, the whole process includes three parts. First, part A is the analysis stage of the system before trimming, in which the functional characteristics of the system to be trimmed are analyzed. Second, part B formulates the trimming plan and searches for the best-matching biological prototype according to its functional characteristics. Finally, part C involves the innovative design stage of the trimmed system.

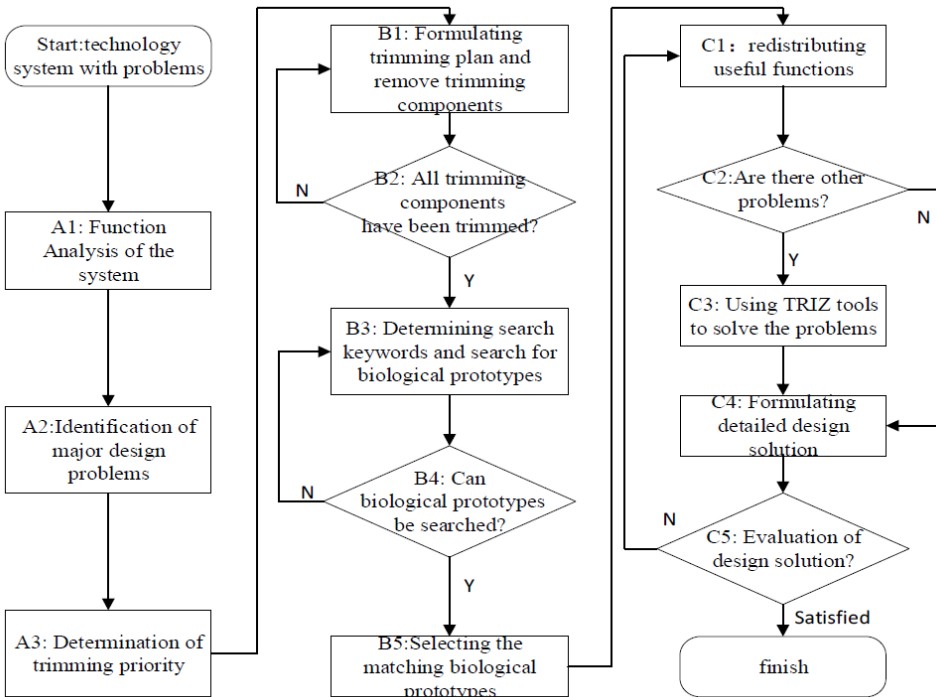

**Figure 1.** Methodology of the trimming-based biologically inspired innovative design.

### 3.1. Analysis before Trimming the System

The analysis performed before trimming the system involves several specific steps to help designers make wise decisions when formulating the trimming plan based on the function-oriented analysis of the product system. This process comprises three specific steps.

Step A1: Analyzing system functions.

Function analysis mainly relies on functional decomposition to analyze the known product by revealing the causal relations between all function carriers within the product system. Each function of the product system plays one of four roles, depending on how it affects the ideality of the system: a useful function is able to increase the ideality of the system; a harmful function leads to a decrease in system ideality; an insufficient function produces lower ideality than required; an excessive function reduces the ideality level of the system, even though its useful effect exceeds requirements. This paper uses a method from a previous study [13] to calculate scores of the function levels of components. For example, "5" is used to indicate a useful function, "−5" denotes a harmful function, "3" is an insufficient function, and "−3" indicates an excessive function.

Step A2: Identifying the main design problem.

The previous step defines harmful and excessive functions in the product system. Subsequently, this step applies CECA to analyze and identify the main design problem in the product system. CECA reveals the causal relations between design problems and physical components that serve as function carriers in the product system with harmful, excessive or inefficient functions. Figure 2 illustrates how CECA is used to identify design problems of a water heater with two causal relations, that is, "or"/"and". The causality between components and functions is established based on CECA. In Figure 2, the causal relations between two components and harmful, excessive and insufficient functions are represented by "or"/"and", which are the premise for calculating the function values for components in the product system.

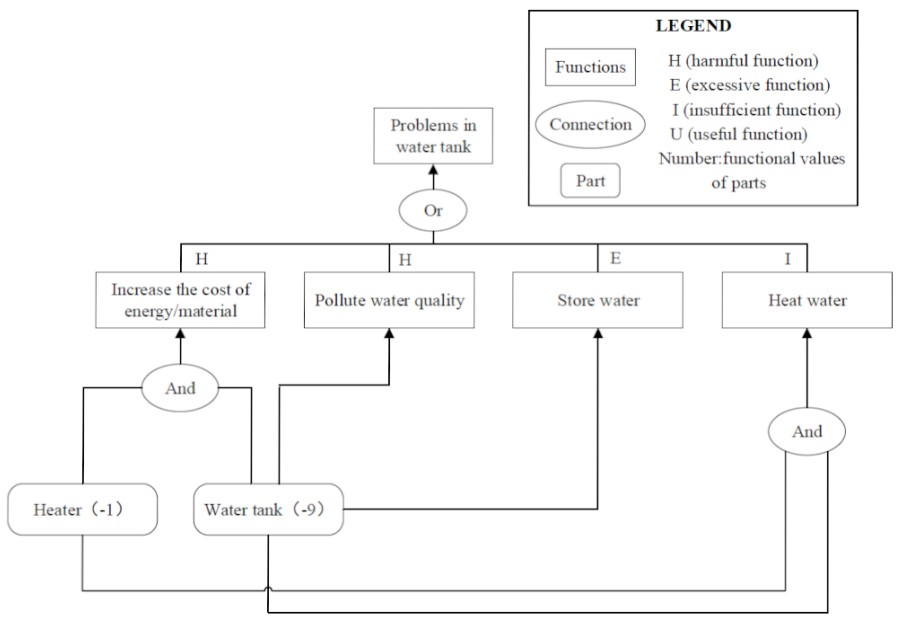

**Figure 2.** Function tree of a water heater product design problem based on CECA.

Step A3: Determining trimming priorities.

After CECA has revealed causal relations between product components and their functional roles in the product system, the function value for every component in the product can be calculated based on weights according to their roles. CECA includes approaches

to tackling "or"/"and" relations. Specifically, each factor in the "and" connection has an equivalent weight in the function unit. For example, in the illustrative case shown in Figure 2, the heater and water tank components work together through the "and" relation and have the harmful function of "increase the costs of energy/material" with a score of "−5 ", so the functional values of the heater and water tank are "−2.5".

$$V_i = \sum_{i=1}^{n} F_i \cdot L_i \cdot T_i \tag{1}$$

Formula (1) represents the calculation of the system component's function value based on the results of CECA. In Formula (1), $V_i$ denotes the value for component $i$, $F_i$ is the function weight, $L_i$ is the weight of the causal relation in CECA, and the value of $T_i$ is assigned by the position of the function in the whole function tree in CECA. The values of harmful and excessive functions located at the root of the CECA-based system function tree are higher than those located at the top because these types of functions are the foundation of the whole causal chain that leads to design problems. For the section containing useful and insufficient functions, the top-level useful function has a lower function value because it is directly related to the requirements of the system.

The case of the water heater in Figure 2 is used as a simple example to explain how to calculate the function value of a product component based on Formula (1). From the CECA result, there is no function tree, and the value of $T_i$ for each function unit is therefore equal to "1". The design problem of the water heater is related to four functions, which comprise two harmful functions, one excessive function and one insufficient function, so the set of $F_i$ can be determined. The harmful function "high cost of energy and material" and the insufficient function "heat water" have the "and" causal relation between their physical carriers; therefore, both $L_i$ values should be "0.5" because the two components contribute equally to the "and" relation. Finally, the weight value of the water tank is denoted by $V_1 = -2.5 - 5 - 3 + 1.5 = -9$, while the heater value is $V_2 = -2.5 + 1.5 = -1$. It is evident that the water tank component, which has a score of "−9", has a lower function value than the heater, which has a value of "−1". Therefore, the water tank is prioritized for trimming.

The main aim of trimming is to improve the ideality of the product system. In this step, according to the function values of the system components calculated in the previous step, the trimming priorities are determined and ranked from low to high. For example, the component with the lowest score in step A3 is the most suitable candidate for elimination to improve the ideality level of the product system. Additionally, the value of the water tank is lower than that of the heater. Therefore, the water tank ranks above the heater in the list of items to be trimmed.

### 3.2. System Trimming Analysis and System Analysis after Trimming

Trimming removes the physical carriers of useful functions when the components with the lowest function values are eliminated. Thus, a system analysis is required after trimming to resolve these new problems. The key objective of this step is to identify resources to carry out the required function. In this study, we extended the searchable range of potential solutions by incorporating BID, which involves five specific steps, as detailed in this section.

Step B1: Formulating the trimming plan and conducting trimming.

The formulation rules of the trimming plan are determined according to the trimming priorities. First, the components with the lowest function values are removed; more than one component may need to be eliminated from the actual system. The identified components are then trimmed according to the formulated trimming plan.

Step B2: Verifying that the identified components have been trimmed.

If there are still components to be trimmed, then the workflow returns to step B1 to reformulate a new trimming plan. If all identified components have been trimmed, then the

completeness of the system after trimming is assessed. The completeness mainly reflects whether all functional requirements are met. If the system is incomplete after trimming, then it is necessary to compensate for the missing function by searching for solutions with similar principles in the knowledge database.

Step B3: Determining search keywords and searching for biological prototypes.

As with other domains, biology is able to provide appropriate principles to solve problems in the system after trimming. The existing biological knowledge database of BID methods is able to provide useful solutions to design problems. In this study, we used the database in MBE [11], in which biological prototypes are graphically represented as functional models that are familiar to engineering designers. Biological prototypes are classified by their functional features in the database. Therefore, useful functions that have been trimmed are used as keywords to search for feasible biological prototypes. During the search for solutions, taxonomy (such as an engineering-to-biology thesaurus [43]) can help to overcome barriers in terminology between engineering and biology. The steps to determine the search keywords are as follows:

(1)    Search for target functions. These are useful functions that are missing from the trimmed system.
(2)    Identify attributes of the target function to describe its characteristics. The target function is described by terms in the form of "V + N" and "V + O + P" (where "V" and "N" refer to specific verbs and nouns, respectively, while "O" and "P" represent abstract objects and attributes).
(3)    In order to improve the adaptability and the degree of structural matching between the identified biological prototype and the design system, the effects and features of the missing target functions are expanded according to the associated components removed from the original design system. The search mode "VOP + the effect/features of the associated function" is used for the main keywords.

Figure 3 shows the search process for biological prototypes. According to the useful functions missing from the trimmed system, the search keywords are constructed, and the keywords are input into the MBE database to search for biological prototypes that can provide the functions required by the trimmed system.

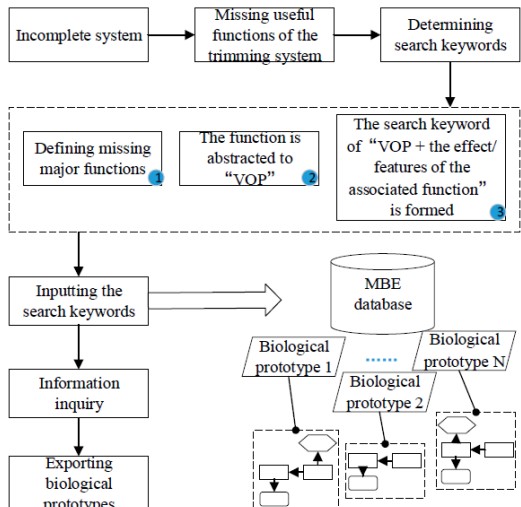

**Figure 3.** The search process for biological prototypes.

Step B4: Verifying that biological prototypes can be identified in the search.

If a biological prototype that matches the keywords is not retrieved from the MBE database, then the composition of the keywords should be changed. First, the missing useful functions are sorted. The function with the lowest function value should be trimmed,

and the search keywords should be determined by the method in step B3 for the remaining functions. If there are still no biological prototypes, then step B4 is repeated until a matching biological prototype is found.

Step B5: Selecting the best-matching biological prototype.

In most circumstances, more than one biological prototype will be found through keyword searching. To identify the best-matching one, a quantitative analysis is performed to determine the similarity between biological prototypes and trimmed useful functions of the system. The biological prototype with the highest similarity is selected as the best match.

In the proposed method, a fuzzy comprehensive evaluation method [50,51] is used to analyze the similarity between the functions of all biological prototypes and the missing useful functions of the design system from three dimensions: compatibility, completeness and feasibility. The following steps constitute the fuzzy comprehensive evaluation method:

B5.1: Establishing the factor set of the comprehensive evaluation.

$$U = \{compatibility, completeness, feasibility\}$$

B5.2: Determining the weight of each factor and establishing the weight vector set.

Compatibility, including functional and flow compatibility, is an important index for evaluating the matched biological prototypes. With the functional models proposed by the database, the biological knowledge in MBE can be easily analyzed by engineering users to determine their functional characteristics. It is suggested that designers first examine whether all functional requirements can be met by the matched biological prototypes before analyzing their completeness and feasibility.

Completeness literally means that the biological system can serve the required function independently without supplementary systems. Generally speaking, the technological implementation of a biological prototype often needs to be supplemented to ensure its workability. However, adding more supplementary systems increases the cost of the redesign. Therefore, the completeness of the biological prototype is evaluated to assess its ideality.

Feasibility is another important indicator of the quality of a biological prototype, and it covers several specific aspects: design constraints, usage context information and efficiency. A biological system usually requires specific conditions for its survival. However, engineering situations are usually too harsh for biological prototypes to work normally. The feasibility of a biological prototype can be evaluated by establishing whether the biological system can function normally under the working conditions of the product.

The overall evaluation of matching biological prototypes is comprehensively decided by all three indicators. If a biological prototype cannot pass the feasibility evaluation, it is directly removed from the solution space. Compatibility and completeness are two indicators for ranking biological solutions by their performance versus their building costs. Five scholars were invited to assign weights to compatibility, completeness and feasibility according to the above analysis. The average scores were calculated, and the final weights were 0.4, 0.2 and 0.4, respectively. Therefore, the weight vector set was $W = \{w1, w2, w3\} = \{0.4, 0.2, 0.4\}$.

B5.3: Carrying out single-factor fuzzy evaluation and establishing the fuzzy evaluation matrix.

The membership set, denoted by $R = \{R_1, R_2, R_3\}$, is determined in this step.

In the compatibility dimension, Formula (2) is used to calculate the similarity, and $R_1 = \{R_{11}, R_{12}\}$ is obtained:

$$CB_{SIM} = \frac{|BP_{\text{sub-f}} \cap DS_{\text{mu-f}}|}{|BP_{\text{sub-f}}| + |DS_{\text{mu-f}}| - |BP_{\text{sub-f}} \cap DS_{\text{mu-f}}|} * \sum_{i=1}^{L} \beta_i \text{sim}(sf_i) \qquad (2)$$

where $CB_{SIM}$ is the similarity between different sub-functions of a biological prototype and the useful functions missing from the design system; $BP$ is the biological prototype; $DS$ is the design system; $BP_{sub-f} \cap DS_{mu-f}$ is the number of similar or identical functions between different sub-functions of the biological prototype and useful functions missing from the design system, and it is denoted by $L$. $BP_{sub-f}$ is the number of sub-functions in the biological prototype, and $DS_{mu-f}$ is the number of sub-functions in the biological prototype, and is the number of useful functions missing from the design system. $\beta_i$ refers to the weights of similar functions i in the functional model of the biological prototype, and $\text{sim}(sf_i)$ refers to the similarity degree between two similar functions among different sub-functions of the biological prototype and useful functions missing from the design system.

The value of $\beta_i$ is determined by the score of the functional model in the system. The weight of a function is determined using a function grade analysis, in which the function grade is defined by the following rules [18]:

(1) If it acts directly on the product, then the function grade is defined as the basic function, and its grade is $B$;
(2) If it acts on the basic function, its grade is $A1$;
(3) Functions that act on grade $i-1$ functions are $Ai$;
(4) The function of the super-system is rated $A1$.

The calculation rules of the function grade are as follows:

(1) The function with the lowest function grade has a value equal to 1;
(2) $Rank(Ai_{-1}) = Rank(A_i) + 1$;
(3) $Rank(B) = Rank(A1) + 2$;
(4) For functions that act on multiple functional components, the rank is the sum of all functions.

The designer calculates the corresponding function grade values in the functional model of the biological prototype and normalizes the values as function weights $\beta_i$, as shown in Formula (3).

$$\beta_i = \frac{x_i - \min}{\max - \min} \tag{3}$$

A function grade diagram of a biological prototype is established, as shown in Figure 4. The function grade and weight values of the biological prototype are shown in Table 1.

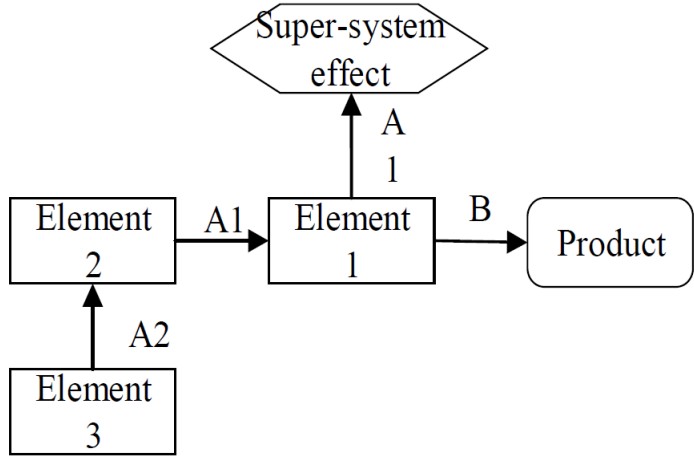

**Figure 4.** Function grade diagram of a biological prototype.

**Table 1.** Function grade and weight values of a biological prototype.

| Name of the Element | Function Grade Values $x_i$ | Weight Values $\beta_i$ (Normalized Results) |
|---|---|---|
| Function 1 | $x_1$ | $\beta_1$ |
| Function 2 | $x_2$ | $\beta_2$ |
| Function 3 | $x_3$ | $\beta_3$ |
| ... | ... | ... |
| Function n | $x_n$ | $\beta_n$ |

The feature similarity analysis and the evaluation of similar functions determine the similarity of function i. The value of the similarity degree $\text{sim}(\text{sf}_i)$ between similar functions is shown in Table 2.

**Table 2.** The values of $\text{sim}(\text{sf}_i)$.

| The Value of $\text{sim}(\text{sf}_i)$ | Meaning |
|---|---|
| 0.2 | Two similar functions are generally not similar |
| 0.4 | Two similar functions are slightly similar |
| 0.6 | Two similar functions are relatively similar |
| 0.8 | Two similar functions are very similar |
| 1 | Two similar functions are completely identical |

In this study, AHP [52] (Analytic Hierarchy Process) was used to determine the membership degree of each function in the completeness and feasibility dimensions. The classification is shown in Table 3. Among the listed factors, completeness can be divided into independent and non-independent execution, which defines whether the biological prototype can independently perform the required functions. Feasibility is classified by whether the biological prototype can perform normally under the working conditions of the product, which can be divided into normal, partially normal and non-normal functioning.

**Table 3.** Evaluation table of completeness and feasibility.

| Factor | Evaluation Level | | |
|---|---|---|---|
| Completeness | Independent execution 1 | | Non-independent execution 0 |
| Feasibility | Normal function 1 | Partially normal function 0.5 | Non-normal function 0 |

According to the above three steps, the fuzzy comprehensive evaluation matrix $R = \begin{bmatrix} R_{11} & R_{12} \\ R_{21} & R_{22} \\ R_{31} & R_{32} \end{bmatrix}$ is obtained.

B5.4: Performing fuzzy comprehensive evaluation.

The membership degree of similar targets is

$$\text{B} = \text{W} \times \text{R} \tag{4}$$

The total similarity calculation formula is

$$SIM(BP) = w1 \times R1 + w2 \times R2 + w3 \times R3 \tag{5}$$

Finally, the biological prototype with the highest total similarity is selected as the best match for the next step of redesigning the trimmed system.

### 3.3. Innovative Design of the System

Once the most suitable biological prototype has been determined, the following steps are followed to redesign the trimmed product. The main purpose of this process is to solve technological problems by merging biological strategies with the product system to overcome their incompatibility. This process comprises four specific steps.

Step C1: Redistributing useful functions of the system after trimming.

The MBE database serves as a source of information on multi-biological effects for designers, but the transformation process between the biological prototype and system design scheme still needs to be solved by designers according to their own experience, so it does not fundamentally improve the practicability of using biological knowledge in product innovative design. In order to resolve the issue of transformation from an existing biological prototype to a system design scheme, this paper proposes a process to derive and construct available resources based on structure–function–attribute analogies, as shown in Figure 5. The retrieved biological knowledge source is used to derive the available resources of the system, thus redistributing the useful functions. The specific process is shown in Figure 6, which can be divided into the following three steps.

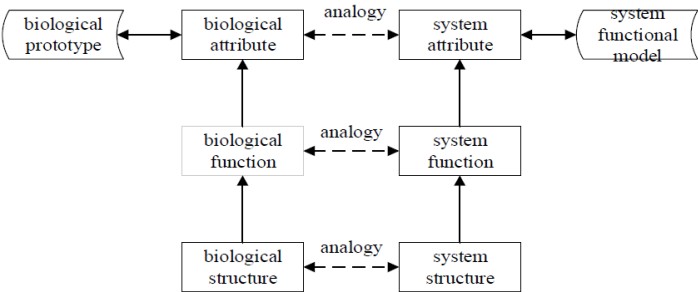

**Figure 5.** Available resource derivation and construction process based on structure–function–attribute analogies.

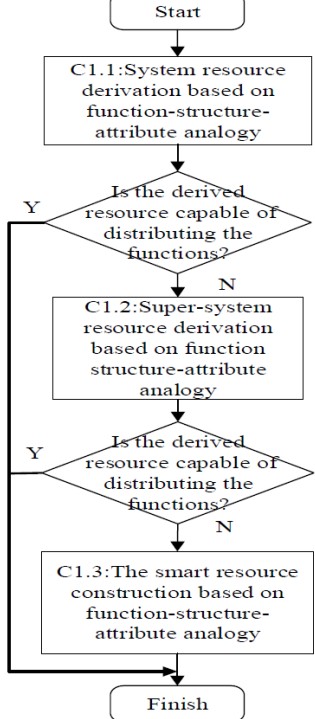

**Figure 6.** The process of redistributing useful functions of the BID-based system.

C1.1: Deriving resources in the system based on structure–function–attribute analogies.

Based on the functional model of the system and the biological prototype, relationships between a system and biological components are established through attribute–component–function analogy mapping. Mapping serves to verify whether the components, functions and attributes of the system are consistent with those provided by the biological prototype. If they are inconsistent, then the available resources in the system can be derived according to the missing attribute characteristics of the system (here, the missing attribute refers to the function and attribute characteristics of the biological prototype compared with those of the remaining components in the system after eliminating components and the missing component or attribute characteristics in the system).

C1.2: Deriving super-system resources based on structure–function–attribute analogies.

If the derived resources in the system are not sufficient to allocate all useful functions, then the same method is used to derive available resources in the super-system based on the missing attribute characteristics of the system.

C1.3: Reconstructing available resources based on structure–function–attribute analogies.

If the derived resources in the system and super-system are still insufficient to allocate all useful functions, then the search continues, and resources outside the system are introduced to reconstruct available resources of the system based on the its missing attribute characteristics.

Step C2: Verifying that all design problems are solved.

After the system is redesigned, it is necessary to verify that no problems remain unsolved. If the system no longer has problems, then the designer continues directly to step C4; otherwise, the next step is employed to solve any remaining problems.

Step C3: Using TRIZ tools to solve remaining problems.

If problems still exist in the system, then TRIZ tools or methods, such as a contradiction matrix, invention principles or 76 standard solutions, are needed to formulate inventive schemes. If there are no problems, the next step is implemented.

Step C4: Formulating design solutions.

If all detected problems have been solved, then a complete design concept will be formulated to enhance the ideality of the design solution.

Step C5: Evaluating the design solution.

The last step is the evaluation of the final design solution to verify its quality. In this step, the design scheme obtained from the trimming process and BID is compared with other design schemes. The aim of the comparison is to verify the effectiveness of the design scheme after applying the BID-based trimming method. Since this is the solution stage of the trimming problem, after the problem is solved, the weighted expert evaluation method is applied to evaluate the ideality of the design solution. The evaluation process is shown in Figure 7, and the specific steps are as follows.

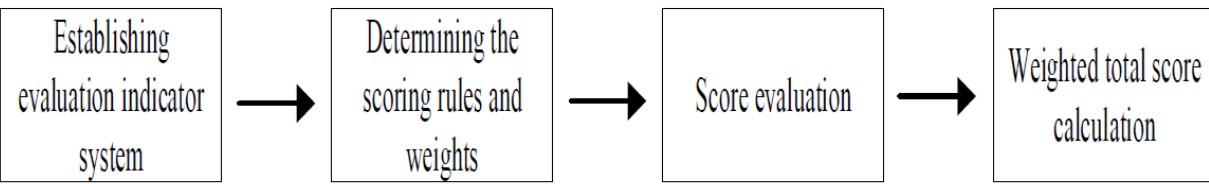

**Figure 7.** Process of the expert evaluation method.

(1)   Establishing an evaluation indicator system.

In this paper, ideality is the main indicator for evaluating the new system compared with the original design. The ideality level formula [53] is

$$Ideality = \sum Benefits / \left( \sum Expenses + \sum Harms \right) \tag{6}$$

In Formula (6), *Ideality* represents the ideality level, *Benefits* represents all useful characteristics and utilities, *Expenses* represents the cost of the whole life cycle, and *Harms* represents harmful effects and undesirable results.

According to Formula (6), the ideality of the system is mainly determined by three indicators: benefits, expenses and harms. Therefore, these three indicators are taken as a comprehensive indicator layer. In a practical situation, the subset of each indicator in the comprehensive indicator layer also contains the corresponding indicator, that is, the project index layer. The three comprehensive indicators "benefit", "expenses" and "harms" are subdivided to establish the final evaluation indicator system, as shown in Table 4, according to the specific meaning of each comprehensive indicator, a reference [54] on mechanical system performance evaluation indicators and a reference [55] on dividing the ideality of a product-service system.

**Table 4.** Evaluation index system of system ideality.

| Target Layer | Comprehensive Indicator Layer (Weight Value) | Project Indicator Layer (Weight Value) |
|---|---|---|
| System ideality | U1 benefits (0.66) | U11 Increasing the usefulness of functions (0.45)<br>U12 Increasing the number of useful functions (0.09)<br>U13 Improving Product Performance (0.30)<br>U14 Increasing productivity (0.16) |
| | U2 costs (0.11) | U21 Design cost (0.10)<br>U22 Production cost (0.30)<br>U23 Cost of ancillary facilities (0.16)<br>U24 Maintenance cost (0.44) |
| | U3 harms (0.23) | U31 Reducing the degree of harmful functions (0.14)<br>U32 Reducing the number of harmful functions (0.08)<br>U33 Existing pollution (0.53)<br>U34 Existing risk (0.25) |

(2)   Determining the evaluation rules and weights.

First, the evaluation rules of the evaluation indicators are formulated according to their difference and size, as shown in Table 5.

**Table 5.** Score evaluation.

| Description | Very Good | Good | Moderate | Not Very Good | Not Good | Very Bad |
|---|---|---|---|---|---|---|
| Score | 9–10 | 7–8 | 5–6 | 3–4 | 1–2 | 0–1 |

In the proposed method, the analytic hierarchy process (AHP) is used to determine the weights, and the relative importance of each factor is measured by introducing an appropriate measurement scale. Thus, a judgment matrix (denoted by A, where element $x_{ij}$ in A represents the relative importance of $e_i$ and $e_j$, and $x_{ji} = 1/x_{ij}$) is constructed, and its consistency is evaluated. Through an inspection of the matrix, its eigenvector is calculated as the corresponding weight value. The evaluation indicators are divided into five grades: equally important, slightly important, important, quite important and extremely important. They correspond to scores of 1, 3, 5, 7 and 9, respectively, while 2, 4, 6 and 8 are the corresponding intermediate grades, as shown in Table 6.

**Table 6.** Comparison of the relative importance of evaluation indicators.

| Meaning | $e_i$ and $e_j$ Equally Important | $e_i$ and $e_j$ Slightly Important | $e_i$ and $e_j$ Important | $e_i$ and $e_j$ Quite Important | $e_i$ and $e_j$ Extremely Important |
|---|---|---|---|---|---|
| $x_{ij}$ | 1 | 3 | 5 | 7 | 9 |

The "summation method" is used to calculate the feature vectors [56] of the judgment matrix:

(1) The original judgment matrix $X = (x_{ij})_{n \times k}$ is normalized according to columns, and $X'$ is obtained:

$$x_{ij}' = \frac{x_{ij}}{\sum\limits_{i=1}^{n} x_{ij}} \tag{7}$$

(2) The vector $\omega'$ is calculated:

$$\omega_i' = \sum\limits_{j}^{k} x_{ij}' \tag{8}$$

(3) $\omega'$ is normalized to obtain the feature vectors:

$$\omega_i = \frac{\omega_i}{\sum\limits_{i=1}^{n} \omega_i'} \tag{9}$$

The weight set of the comprehensive indicator layer is determined by the above method and is denoted by $\omega = \{\omega1, \omega2, \omega3\}$, and the weight set of the project indicator layer is $\omega_i = \{\omega_{i1}, \omega_{i2}, \cdots, \omega_{ij}\}$. Due to space limitations, this paper presents the project indicator layer according to the above method and the judgment matrix $A$ of the first layer:

$$\begin{bmatrix} A & U_1 & U_2 & U_3 \\ U_1 & 1 & 7 & 7 \\ U_2 & 1/7 & 1 & 1/3 \\ U_3 & 1/7 & 3 & 1 \end{bmatrix}$$

The matrix passes the consistency test. Then, using Formulas (7)–(9), its feature vector can be obtained: $\omega = \{\omega1, \omega2, \omega3\} = \{0.66, 0.11, 0.23\}$.

Similarly, the second judgment matrix $A1$, $A2$ and $A3$ can be obtained by using the above method:

$$\begin{bmatrix} A_1 & U_{11} & U_{12} & U_{13} & U_{14} \\ U_{11} & 1 & 3 & 3 & 3 \\ U_{12} & 1/3 & 1 & 1/5 & 1/3 \\ U_{13} & 1/3 & 5 & 1 & 3 \\ U_{14} & 1/3 & 3 & 1/3 & 1 \end{bmatrix}$$

$$\begin{bmatrix} A_2 & U_{21} & U_{22} & U_{23} & U_{24} \\ U_{21} & 1 & 1/3 & 1/3 & 1/3 \\ U_{22} & 3 & 1 & 5 & 1/3 \\ U_{23} & 3 & 1/5 & 1 & 1/3 \\ U_{24} & 3 & 3 & 3 & 1 \end{bmatrix}$$

$$\begin{bmatrix} A_3 & U_{31} & U_{32} & U_{33} & U_{34} \\ U_{31} & 1 & 3 & 1/5 & 1/3 \\ U_{32} & 1/3 & 1 & 1/5 & 1/3 \\ U_{33} & 5 & 5 & 1 & 3 \\ U_{34} & 3 & 3 & 1/3 & 1 \end{bmatrix}$$

The matrix above passes the consistency test. Formulas (7)–(9) are used to obtain its feature vector as follows:

$$\omega_1 = \{0.45, 0.09, 0.30, 0.16\}$$
$$\omega_2 = \{0.10, 0.30, 0.16, 0.44\}$$
$$\omega_3 = \{0.14, 0.08, 0.53, 0.25\}$$

The weights of the indicators in the comprehensive and project indicator layers are shown in Table 4.

(4)    Calculating the total weighted score.

The weighted formula is used to calculate the evaluation results:

$$W = \frac{\sum\limits_{1}^{m}(\sum\limits_{i=1}^{n} A_i W_i)}{m} \tag{10}$$

In the formula, $m$ refers to the number of experts, $n$ is the number of evaluation indices, $A_i$ is the score of evaluation index $i$, $W_i$ is the weight of evaluation index $i$, and $W$ is the total score.

The design solution is evaluated according to the above steps and is completed when it has been proven to improve the system ideality. Otherwise, the design solution should be further optimized by returning to step C1; thus, the product system is redesigned until a satisfactory design solution is obtained.

## 4. Case Study

A steel tape armoring machine is used as an example to illustrate how to apply the three key steps of the proposed method using the workflow in Figure 1: system analysis before trimming, system trimming analysis and system analysis after trimming to obtain an innovative system design.

### 4.1. Analysis of the Steel Tape Armoring Machine before Trimming

The metallic armor illustrated in Figure 8a is an indispensable component of a cable that needs to resist stretching. The armoring machine shown in Figure 8b is a current device used to coat cables with steel tape. However, there are some significant problems in the current system. One of the most significant flaws of the current machine is that the roller vibrates extensively as the rotating speed of the steel tape plate increases, resulting in loud noise and harmful effects on the machine. Therefore, in order to maintain an acceptable level of vibration, a limitation is set on the rotating speed of the steel tape plates, which, in turn, reduces the productivity of the armoring machine. This is a clear opportunity for redesigning the high-speed steel tape armoring machine by eliminating the trade-off that causes the noise. Therefore, the proposed methodology was applied to solve this design problem by following the workflow in Figure 1.

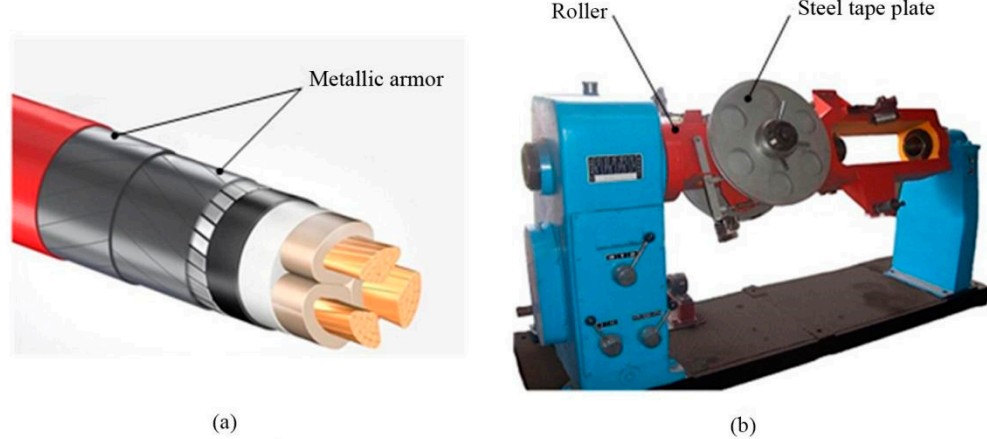

(a)          (b)

**Figure 8.** (**a**) An indispensable component of a cable; (**b**) Illustrations of the cable armoring machine.

Step A1: Performing function analysis of the existing steel tape armoring machine.

One of the most significant causes of vibration in the analyzed device is that the weights of the steel plates decrease as they wind around the cable during the armoring process. It is difficult to maintain the same rate of weight loss for two plates, and the accumulation of weight differences leads to significant vibration and noise. The rotating speed of steel plates in the armoring machine is an important factor that causes a contradiction: on the one hand, the speed should be high since it is closely related to productivity, and on the other hand, the speed should be low because it is the root cause of vibration. In order to eliminate this fundamental problem, a function analysis of the existing product system is first conducted before formulating the trimming plan. The functional model of the existing product is shown in Figure 9.

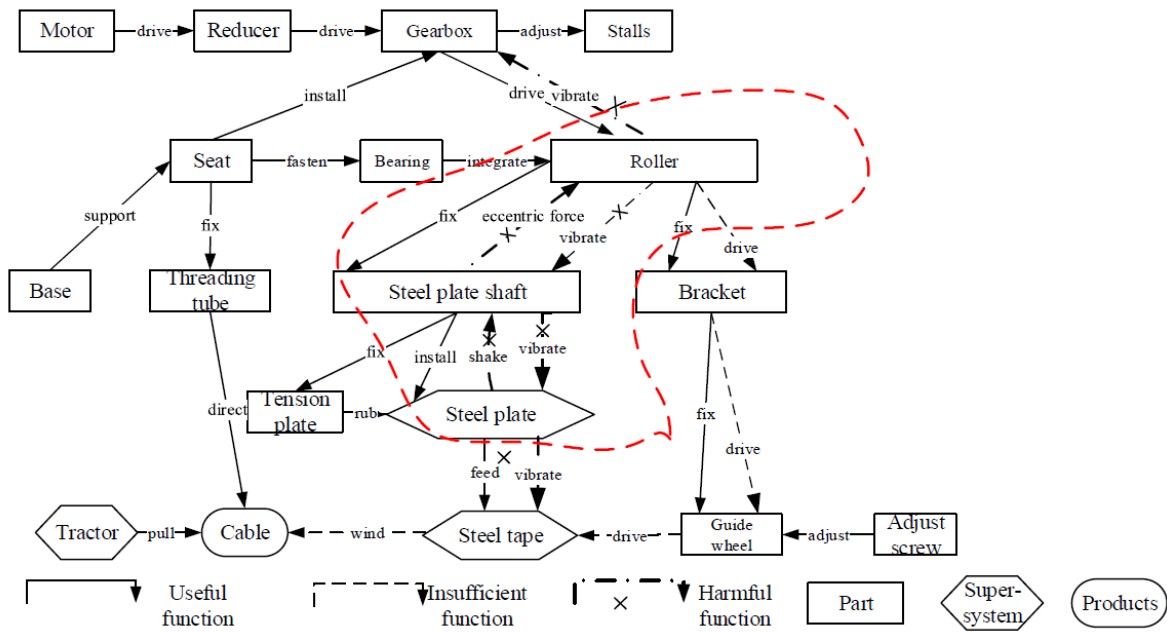

**Figure 9.** Functional model of the original steel tape armoring machine.

Step A2: Identifying major design problems.

Referring to the functional model of the original system, CECA can be performed as shown in Figure 10 to indicate causal relationships between components and the different functional roles that they play. The information in Figure 9 reveals a major harmful

function, i.e., the vibration of the roller. All of the causal relations between the functions and components are represented in Figure 10.

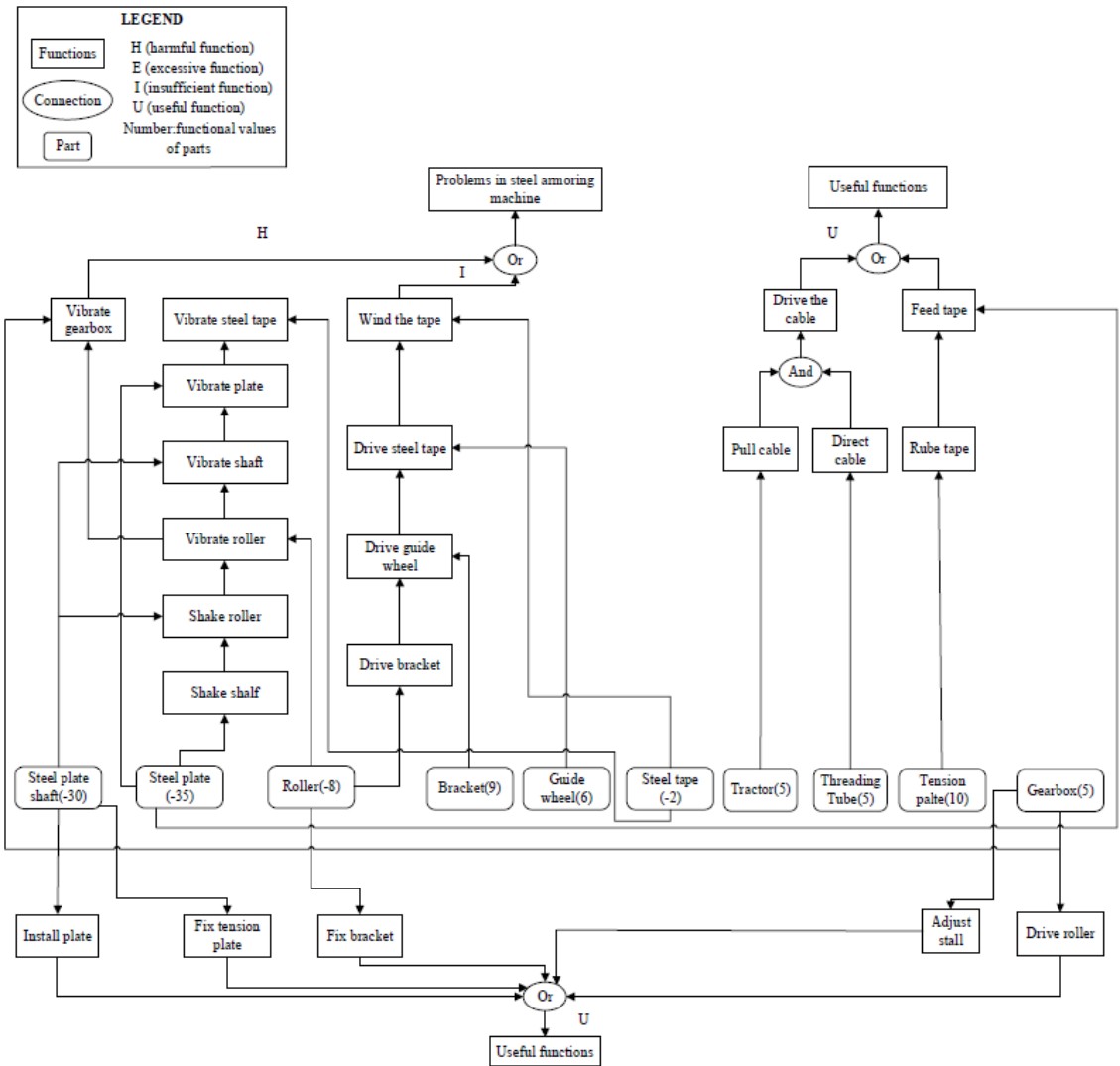

**Figure 10.** Function tree of the original model of the steel tape armoring machine based on CECA (partial).

Step A3: Determining trimming priorities.

Based on the CECA result, the function value for each component can be calculated by Formula (1). The results are listed in Table 7, which includes key variables such as $F_i$, $L_i$ and $T_i$ based on the causal relations between functional roles and components.

All components in Table 7 are ranked by their function values. The ranked results in Table 7 are used to define trimming priorities. As the steel plate shaft, steel plate and roller have negative function values, they are prioritized for trimming.

**Table 7.** Function values for the components in the original armoring machine.

| Component | Useful Function | | | Insufficient Function | | | Excessive Function | | | Harmful Function | | | Function Value | Rank |
|---|---|---|---|---|---|---|---|---|---|---|---|---|---|---|
| | Fi | Li | Ti | Fi | Li | Ti | Fi | Li | Ti | Fi | Li | Ti | | |
| Plate Shaft | 5 | 1 | (1 + 1) | - | - | - | - | - | - | −5 | 1 | (3 + 5) | −30 | 2 |
| Steel plate | 5 | 1 | 1 | - | - | - | - | - | - | −5 | 1 | (2 + 6) | −35 | 1 |
| Roller | 5 | 1 | 1 | 3 | 1 | 4 | - | - | - | −5 | 1 | 5 | −8 | 3 |
| Gearbox | 5 | 1 | 1 + 1 | - | - | - | - | - | - | −5 | 1 | 1 | 5 | 4 |
| Bracket | - | - | - | 3 | 1 | 3 | - | - | - | - | - | - | 9 | 6 |
| Guide wheel | - | - | - | 3 | 1 | 2 | - | - | - | - | - | - | 6 | 5 |
| Steel tape * | - | - | - | 3 | 1 | 1 | - | - | - | −5 | 1 | 1 | −2 | - |
| Tractor * | 5 | 0.5 | 2 | - | - | - | - | - | - | - | - | - | 5 | - |
| Threading tube | 5 | 0.5 | 2 | - | - | - | - | - | - | - | - | - | 5 | 4 |
| Tension plate | 5 | 1 | 2 | - | - | - | - | - | - | - | - | - | 10 | 7 |

* Super-system component.

### 4.2. The Trimming Analysis of the Steel Tape Armoring Machine and the Analysis of the Device after Trimming

Step B1: Formulating the trimming plan and trimming the selected components.

From the results in Table 1, a trimming plan is formulated to remove all three components that have negative function values. If these components are trimmed, the structure of their connected components will change; therefore, components that are connected to the trimmed components must be assessed to determine whether they need to be redesigned as well. Figure 4 highlights the trimmed area, which is circled by a red dotted line.

Step B2: Verifying that the designated components have been trimmed.

First, it is confirmed that the three designated components (steel shaft, steel plate and roller) have been trimmed. An evaluation of the system reveals that the trimmed components were carriers of useful functions, namely, feeding the steel tape, driving the bracket and fixing the bracket, which have been removed from the system. Therefore, in the next step, new components are needed as function carriers to work with the original components.

Steps B3–B4: Determining search keywords and searching for appropriate biological prototypes.

The steps to determine search keywords are followed to analyze the useful functions that are now missing from the steel tape armoring machine due to trimming:

(1) Among the missing useful functions of the steel tape armoring machine determined in the previous step is the target function "feed the steel tape".
(2) The attribute "feed the steel tape" is characterized by the descriptor "feed long strips of solid material".
(3) In order to match the adaptability and structure of the steel tape armoring machine with those of the biological prototype, the functional feature of the "winding cable" function is summarized as "winding solid" according to the functional model of the original design. Therefore, the search keyword phrase is "feeding long strip solid, winding around another object".

After searching the MBE database, two biological prototypes match the required function. The two design cases are "silkworm silks its cocoon" and "spider wind the prey by its silk". The retrieved results are shown in Figure 11a,b, and the functional models of the two biological prototypes are shown in Figure 11c,d. Since the search keywords in this case are successful in identifying biological prototypes, we can proceed directly to the next step to rank the prototypes and select the best match.

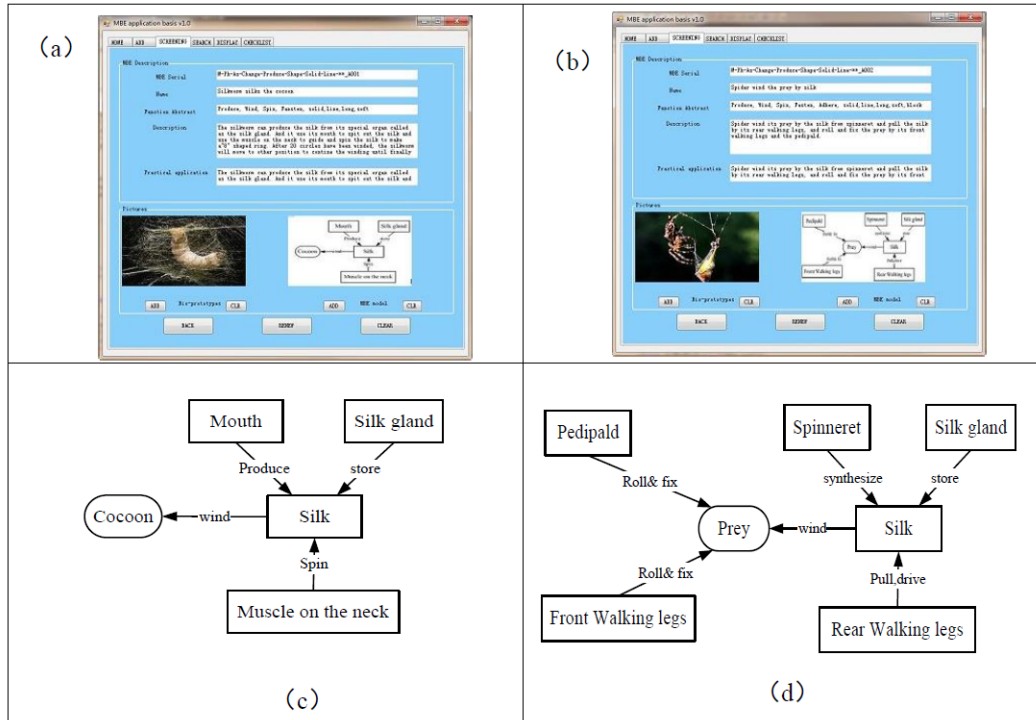

**Figure 11.** (**a**,**b**) Retrieved pages of matching biological prototypes; (**c**,**d**) Functional models of matching biological prototypes.

Step B5: Selecting the best-matching biological prototype.

The similarity between the biological prototype and the target function is calculated from three dimensions: compatibility, completeness and feasibility.

B5.1–B5.2: The factor set is determined in the form $U = \{compatibility, completeness, feasibility\}$, and the weight vector set is $W = [0.4, 0.2, 0.4]$.
B5.3: Establishing the fuzzy evaluation matrix.

In the feasibility dimension, Formula (2) is used to calculate the similarity between the target function and each sub-function of the cocoon and spider. According to the functional model of the biological prototype in Figure 10, the main sub-functions of the spider are: roll fix prey 1, roll fix prey 2, synthesize silk, store silk, pull and drive silk, and wind prey. The main sub-functions of the cocoon are: store silk, spin silk, produce silk and wind cocoon. The functional models of the biological prototypes "spider winds the prey by its silk" and "silkworm silks its cocoon" are analyzed by using function grades. The obtained function grade values and weights of each function are shown in Tables 8 and 9.

**Table 8.** Function grade values and weights of "spider winds the prey by its silk".

| The Names of the Function | Function Grade Values $x_i$ | Weight Values $\beta_i$ (Normalized Results) |
|---|---|---|
| roll fix prey 1 | B–3 | 0.25 |
| roll fix prey 2 | B–3 | 0.25 |
| synthesize silk | A1–1 | 0.08 |
| pull and drive silk | A1–1 | 0.08 |
| store silk | A1–1 | 0.08 |
| wind cocoon | B–3 | 0.25 |

**Table 9.** Function grade values and weights of "silkworm silks its cocoon".

| The Names of the Function | Function Grade Values $x_i$ | Weight Values $\beta_i$ (Normalized Results) |
|---|---|---|
| produce silk | A1–1 | 0.17 |
| store silk | A1–1 | 0.17 |
| spin silk | A1–1 | 0.17 |
| wind cocoon | B–3 | 0.50 |

The similarity degree is obtained from Formula (2):

$$CB_{SIM}(\text{spider}) = \frac{3}{5+3-3} \times (0.08 \times 0.6 + 0.25 \times 0.4 + 0.25 \times 0.4) = 0.15, \ CB_{SIM}(\text{cocoon}) = \frac{1}{4+3-1} \times 0.17 \times 0.6 = 0.02$$

The result of this calculation is $R_1 = [0.15, 0.02]$.

In the completeness dimension, the spider biological prototype has three independent components that can realize the required function, one of which is "rear walking legs", which can be applied to "feed" the material. The other two components equally fulfill the function of "roll" the material. However, the "silkworm" solution does not cover the required function of "roll or fix" the object to be wound by the material since the silk cocoon is an accumulation of only the silk itself. Therefore, the completeness of the prototype "spider winds its prey" is higher than that of the "silkworm" prototype. Thus, $R_2 = [1, 0]$.

In the feasibility dimension, since both cocoons and spiders can function normally under the working conditions of an armoring machine, $R_3 = [1, 1]$.

From the above analysis, the fuzzy evaluation matrix is R $= \begin{bmatrix} 0.15 & 0.02 \\ 1 & 0 \\ 1 & 1 \end{bmatrix}$.

B5.4: Performing the fuzzy comprehensive evaluation.

The total similarity values for the spider and the cocoon were calculated using Formula (5):

$$SIM(spider) = 0.4 \times 0.15 + 0.2 \times 1 + 0.4 \times 1 = 0.66, SIM(cocoon) = 0.4 \times 0.02 + 0.2 \times 0 + 0.4 \times 1 = 0.41$$

Thus, B = W × R = $[0.66, 0.41]$. Therefore, the "spider winds its prey" is chosen as the prototype for redesigning the product.

*4.3. Innovative Design of the Steel Tape Armoring Machine*

Step C1: Redistributing useful functions of the system after trimming.

Using the retrieved biological prototype, the available resources for the steel tape armoring machine after trimming are derived, and thus, the useful functions of the system are reallocated.

C1.1: Deriving resources within the system based on structure–function–component analogies.

In order to reallocate the missing useful functions of the steel tape armoring machine, the resources available in the system are first determined. In a structure and function analogy, the steel tape can be regarded as the spider silk, and the cable can be considered the prey. The process of the spider winding silk around its prey is biologically analogous to applying the steel tape armor to the cable. BID guides the technological transformation of the biological prototype; in other words, the functional model is a blueprint from which the designer can determine the technological measures to replace components.

Figure 12 depicts the analogical relations between the functions of the steel tape armoring machine and those of the biological prototype (the spider winding its prey with silk). For example, in the armoring machine, the steel tape plays the role of the silk, and the cable plays the role of the prey. The most significant difference between the technological and biological systems is the sequence in which the material is wound around the object: in the biological system, it is realized by two independent actions, while in the armoring

machine, the winding action is the combined result of pulling the cable and rolling the steel tape.

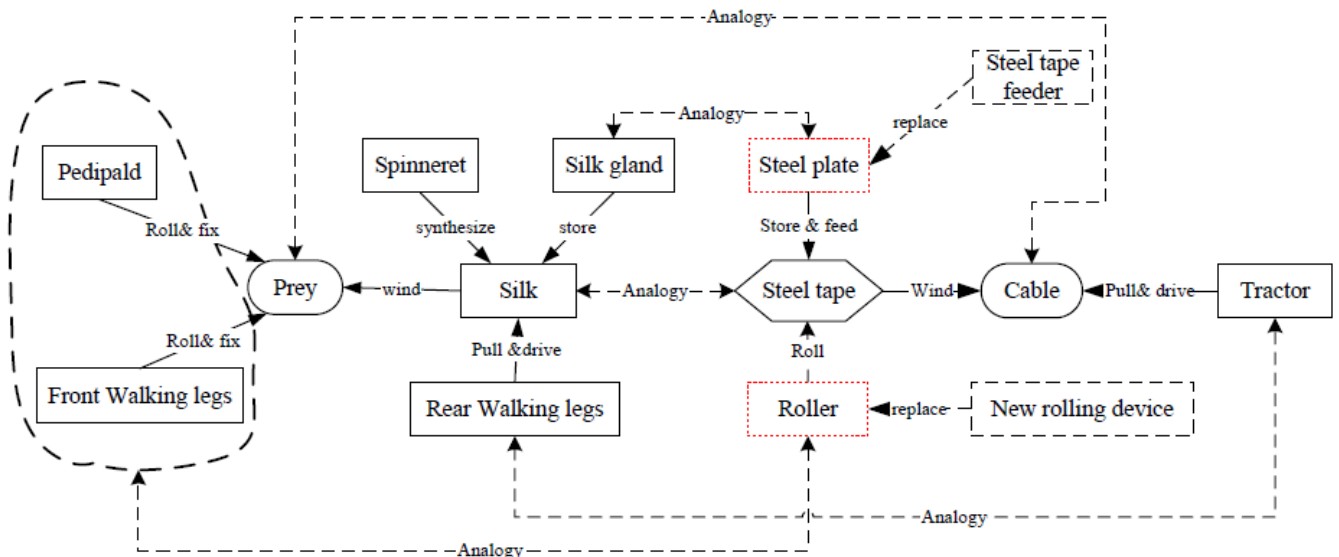

**Figure 12.** Analogical analysis for rebuilding the product system based on the biological prototype.

In the post-trimming product system, the tractor serves the function of pulling the cable, but since the original roller has been removed, the steel tape needs to be rolled by a new component. As a technological solution, a new component is installed at the base of the armoring machine to feed the steel tape, and a new rolling bracket is designed to drive and roll the steel tape onto the cable. As the resources in the system derived from structure-function-component analogies can provide the useful functions missing from the trimmed system, the functional requirements are met. Thus, we can proceed straight to step C2 to verify that there are no other problems in the system.

Step C2: Verifying that there are no other design problems in the steel tape armoring machine.

The redesign of the armoring machine has eliminated the harmful function of vibration and improved its productivity. However, the potential danger posed to people around the device is still unresolved in the redesigned solution. This problem is caused by the improved winding speed, which can produce a strong centrifugal force that results in the fracture of the steel tape. If the steel tape is broken while rotating at high speeds, it poses significant danger to people nearby.

Step C3: Using TRIZ tools to solve problems.

The potential danger in the redesigned system can be addressed by applying the 76 standard solutions of TRIZ. This design problem falls into class 1.2, "eliminating or neutralizing harmful effects". Solution No. 1.2.3 is applicable to problems in which the harmful action is caused by a field and entails introducing an element (S3 in Figure 13) to absorb the harmful field and thereby eliminate it. In this case, a protective shell is designed as the object (denoted by "S3" in Figure 13) to absorb the harmful field caused by the broken steel tape. The design process is shown in Figure 13.

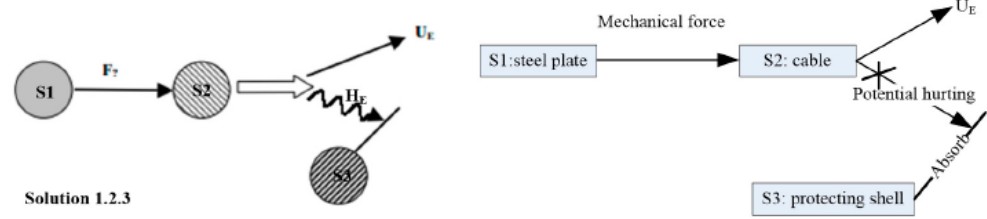

**Figure 13.** Process of designing a protective shell to eliminate the harmful effect.

Step C4: Formulating the design solution.

The final design of the conceptual solution is shown in Figure 14, and its systematic model is shown in Figure 15.

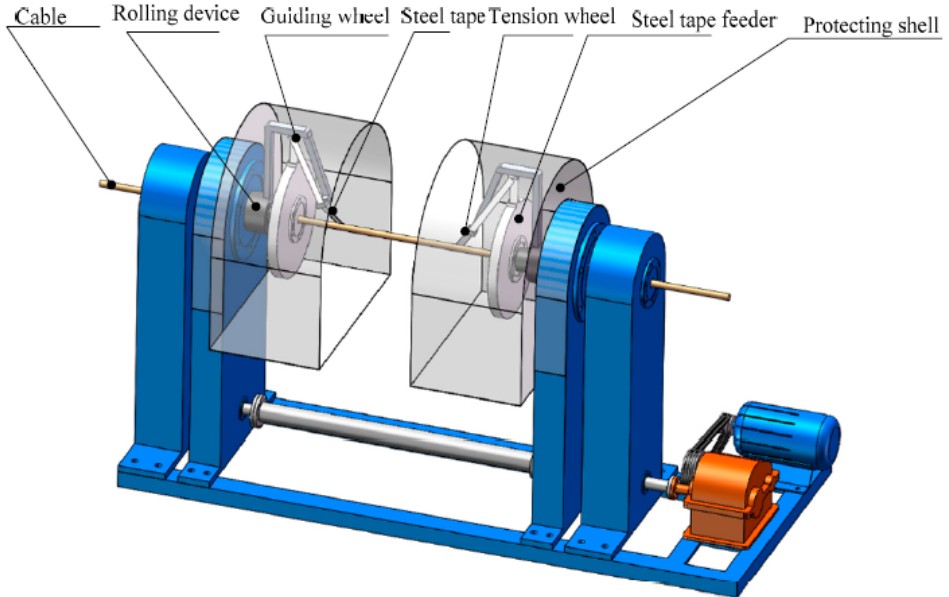

**Figure 14.** Conceptual solution of the refined cable armoring machine.

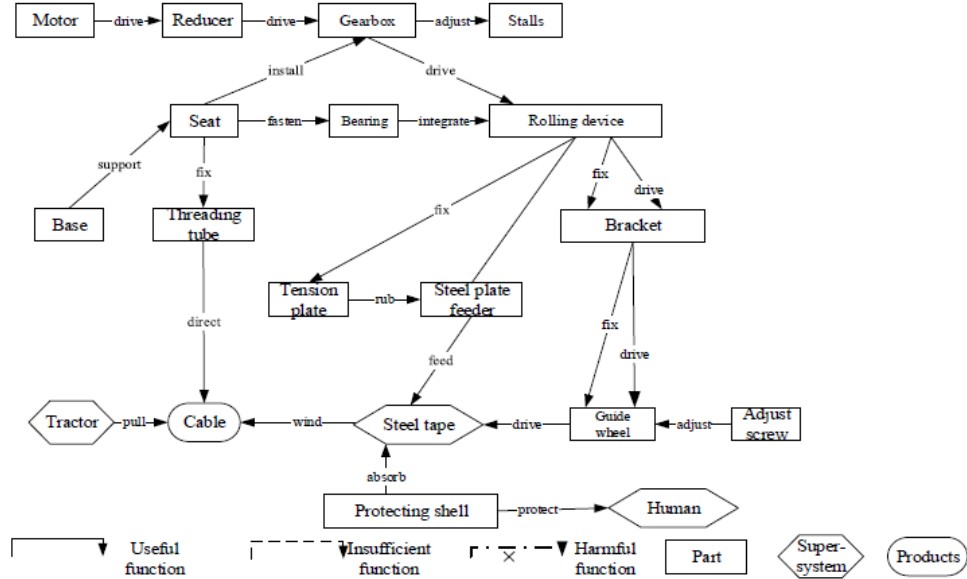

**Figure 15.** Systematic model of the redesigned machine.

Step C5: Evaluating the design solution.

In the following evaluation, the redesigned steel tape armoring machine is scheme 1, and the original steel tape armoring machine is scheme 2. A relevant search was carried out to find other design schemes of the steel tape armoring machine, and the results are shown in Table 10.

**Table 10.** Descriptions of other design options.

| Design Scheme | Scheme 3 [57] | Scheme 4 | Scheme 5 |
|---|---|---|---|
| Description of the Scheme | A steel tape armoring machine containing brackets, a guide wheel, a steel tape plate, steel tape, a cable, a roller and other structures. The steel tape plate traverses the roller, which can eliminate the influence of vibration and noise. The roller performs the function of the steel tape plate shaft to provide support and rotation. Its advantages include simple structure, high production efficiency, low noise, etc. | A self-interlocking armoring machine for submersible pump cables, including the main body of the armoring machine, the inlet frame, the outlet frame, the guide wheel, the limiting pulley, the active mixing wheel and the limiting ring. It solves the problems of cable wear and non-adjustable cable height due to the different import heights of cables and devices. | An interlocking armoring machine that includes a bottom plate, support column, top cover, fixed ring, support pipe, storage plate, installation frame, rotary motor, transmission belt, fixed turntable, fixed crossbar, auxiliary ring and other structures. The top cover can be opened and closed, which is convenient for installation, disassembling and maintenance. The limit wheel limits the conveyor belt to effectively prevent it from falling off. The machine is characterized by reduced labor, quick replacement, high efficiency, and a simple and stable structure. |
| Picture of the Scheme |  |  |  |

Scheme 4 and Scheme 5 are provided by Patent CN212084745U(China) and CN209087462U(China) respectively.

In this study, experts, including engineering, technical, manufacturing, maintenance and front-line technical personnel, were invited to form an evaluation group. According to the established scoring standards, each indicator was scored for both the original and new designs of the steel tape armoring machine and for steel tape armoring machine designs obtained by other methods. Seven reviewers were invited to score each indicator in the project indicator layer (see the Appendix A for detailed scoring results), and the scoring results were inserted into Formula (10) to calculate the final system ideality, as shown in Table 11. According to the final evaluation results shown in Table 9, the ideality of the steel tape armoring machine designed by the BID-based trimming method is significantly improved.

**Table 11.** Final evaluation results of system ideality.

| Comprehensive Evaluation Indicator | Scheme 1 | Scheme 2 | Scheme 3 [57] | Scheme 4 | Scheme 5 |
|---|---|---|---|---|---|
| Benefits | 6.64 | 4.85 | 5.89 | 5.08 | 5.87 |
| Expenses | 5.24 | 5.46 | 5.57 | 5.41 | 5.27 |
| Harms | 5.46 | 6.37 | 5.28 | 5.86 | 5.61 |
| The final score for ideality | 6.34 | 5.27 | 5.71 | 5.30 | 5.74 |

Scheme 4 and Scheme 5 are provided by Patent CN212084745U (China) and CN209087462U (China) respectively.

According to the above evaluation results, we can conclude that the proposed trimming method based on BID is highly effective in generating innovation compared with other methods. The proposed approach can facilitate the development of innovative solutions by designers and help them produce novel design schemes. Such outcomes are supported by previous findings in the literature [58].

## 5. Conclusions

This paper contributes a methodology that integrates trimming with BID, which can facilitate the application of biological solutions to inventive engineering problems. With the proposed method, biological strategies are applied to rebuild post-trimmed products. Moreover, the trimming process can provide a suitable scenario for exploring and exploiting biological solutions in engineering designs. From a broad perspective, the proposed method represents a new function-oriented attempt to integrate trimming and BID for innovation.

Despite the advantages of the new design method, this study also has some limitations. First, although it is laborious to graphically represent functional models of biological prototypes, it is an indispensable step in the proposed method, as it bridges the gap between biological and engineering domains. However, in the MBE database applied in this study, only partial biological prototypes can be functionally modeled at present, and the functional models are stored in the MBE database. Second, in order to manage the incompatibility between the biological system and the technological product design, designers have to search for and identify appropriate technological counterparts to generate final product redesign solutions. This requires designers to have certain design knowledge and experience, which, in turn, may affect their ability to apply biological information. Third, it is difficult for most engineering designers to apply biological knowledge from an unfamiliar domain. Although previous studies on BID have provided various tools and methods to help engineering users, configuring the biological prototype in the context of engineering applications largely depends on the skills and experience of designers. Fourth, some steps of the design method proposed in this study may rely on subjective judgment. Finally, only one case was used to verify the feasibility of the design method, which may limit its generalizability.

Four key research directions should be explored in the future to improve the proposed method and overcome the above-mentioned limitations. First, additional approaches to incorporating TRIZ into BID will be conducted with the aim of producing more efficient inventive solutions. Moreover, the biological knowledge database will be continuously developed by adding new biological prototypes, which will strengthen its applicability to more design problems. Then, it will be necessary to develop an intelligent retrieval module to manage the continual expansion of the biological knowledge base. Finally, we expect to make the proposed method more versatile by refining some of the details provided in this article, and the results of these efforts will be verified using more case studies.

**Author Contributions:** Conceptualization, F.Y., and W.L.; project administration, P.Z.; resources, W.L.; supervision, Z.N., and P.Z.; writing—review and editing, X.L., and Z.N.; writing—original draft, P.Z., F.Y., and X.L. All authors have read and agreed to the published version of the manuscript.

**Funding:** This paper was funded by the National Natural Science Foundation of China (Grant No. 51805142), the Natural Science Foundation of Hebei province of China (Grant No. E2017202260) and the Creative Research Groups of the Natural Science Foundation of Hebei province of China (Grant No. E2020202142).

**Conflicts of Interest:** The authors declare that there is no conflict of interest regarding the publication of this paper.

# Appendix A

**Table A1.** Evaluation scores of the comprehensive indicator "Benefits".

| Design Scheme | Project Indicators | Reviewer 1 | Reviewer 2 | Reviewer 3 | Reviewer 4 | Reviewer 5 | Reviewer 6 | Reviewer 7 | The Final Score |
|---|---|---|---|---|---|---|---|---|---|
| Scheme 1 | Increasing the usefulness of functions | 7 | 8 | 7 | 6 | 7 | 6 | 8 | |
| | Increasing the number of useful functions | 5 | 5 | 6 | 5 | 7 | 5 | 6 | |
| | Improving Product Performance | 7 | 6 | 7 | 8 | 6 | 7 | 6 | 6.64 |
| | Increasing productivity | 8 | 7 | 7 | 6 | 7 | 7 | 8 | |
| | Scoring results | 6.29 | 6.97 | 6.91 | 6.37 | 6.70 | 6.37 | 6.90 | |
| Scheme 2 | Increasing the usefulness of functions | 5 | 6 | 5 | 4 | 4 | 6 | 5 | |
| | Increasing the number of useful functions | 4 | 5 | 5 | 4 | 4 | 5 | 4 | |
| | Improving Product Performance | 5 | 4 | 5 | 4 | 5 | 4 | 4 | 4.85 |
| | Increasing productivity | 6 | 5 | 5 | 5 | 6 | 6 | 5 | |
| | Scoring results | 5.07 | 5.15 | 5.00 | 4.16 | 4.62 | 5.31 | 4.61 | |
| Scheme 3 | Increasing the usefulness of functions | 6 | 6 | 7 | 5 | 6 | 7 | 7 | |
| | Increasing the number of useful functions | 5 | 6 | 5 | 4 | 6 | 5 | 5 | |
| | Improving Product Performance | 6 | 5 | 6 | 7 | 5 | 5 | 6 | 5.89 |
| | Increasing productivity | 7 | 5 | 6 | 5 | 6 | 5 | 6 | |
| | Scoring results | 6.07 | 5.54 | 6.36 | 5.51 | 5.70 | 5.90 | 6.36 | |
| Scheme 4 | Increasing the usefulness of functions | 5 | 5 | 6 | 4 | 5 | 5 | 6 | |
| | Increasing the number of useful functions | 5 | 4 | 4 | 4 | 4 | 5 | 4 | |
| | Improving Product Performance | 6 | 5 | 6 | 5 | 6 | 5 | 4 | 5.08 |
| | Increasing productivity | 5 | 4 | 5 | 5 | 5 | 6 | 5 | |
| | Scoring results | 5.30 | 4.75 | 5.66 | 4.46 | 5.21 | 5.61 | 4.61 | |
| Scheme 5 | Increasing the usefulness of functions | 6 | 7 | 6 | 6 | 7 | 6 | 6 | |
| | Increasing the number of useful functions | 5 | 5 | 5 | 4 | 4 | 5 | 6 | |
| | Improving Product Performance | 7 | 6 | 5 | 6 | 5 | 6 | 5 | 5.87 |
| | Increasing productivity | 6 | 6 | 5 | 5 | 6 | 5 | 6 | |
| | Scoring results | 6.21 | 6.36 | 5.45 | 5.66 | 5.97 | 5.75 | 5.70 | |

**Table A2.** Evaluation score of the comprehensive indicator "costs".

| Design Scheme | Project Indicators | Reviewer 1 | Reviewer 2 | Reviewer 3 | Reviewer 4 | Reviewer 5 | Reviewer 6 | Reviewer 7 | The Final Score |
|---|---|---|---|---|---|---|---|---|---|
| Scheme 1 | Design cost | 5 | 6 | 5 | 5 | 5 | 6 | 6 | |
| | Production cost | 5 | 5 | 6 | 5 | 6 | 6 | 5 | |
| | Cost of ancillary facilities | 6 | 5 | 5 | 6 | 5 | 6 | 5 | |
| | Maintenance cost | 5 | 4 | 5 | 6 | 4 | 5 | 6 | 5.24 |
| | Scoring results | 5.16 | 4.66 | 5.30 | 5.60 | 4.86 | 5.56 | 5.54 | |
| Scheme 2 | Design cost | 5 | 5 | 6 | 5 | 6 | 5 | 5 | |
| | Production cost | 6 | 6 | 5 | 6 | 5 | 6 | 5 | |
| | Cost of ancillary facilities | 5 | 5 | 6 | 5 | 5 | 6 | 6 | |
| | Maintenance cost | 5 | 6 | 5 | 5 | 6 | 6 | 5 | 5.46 |
| | Scoring results | 5.30 | 5.74 | 5.26 | 5.30 | 5.54 | 5.90 | 5.16 | |
| Scheme 3 | Design cost | 5 | 6 | 5 | 6 | 6 | 5 | 6 | |
| | Production cost | 6 | 5 | 6 | 5 | 5 | 6 | 6 | |
| | Cost of ancillary facilities | 6 | 6 | 5 | 5 | 6 | 6 | 5 | |
| | Maintenance cost | 6 | 6 | 6 | 5 | 6 | 5 | 6 | 5.57 |
| | Scoring results | 5.90 | 5.70 | 5.74 | 5.10 | 5.70 | 5.46 | 5.40 | |
| Scheme 4 | Design cost | 5 | 5 | 6 | 5 | 6 | 5 | 6 | |
| | Production cost | 6 | 5 | 6 | 6 | 5 | 5 | 5 | |
| | Cost of ancillary facilities | 6 | 6 | 5 | 6 | 6 | 6 | 5 | |
| | Maintenance cost | 5 | 5 | 5 | 5 | 6 | 5 | 6 | 5.41 |
| | Scoring results | 5.46 | 5.10 | 5.40 | 5.46 | 5.70 | 5.16 | 5.54 | |
| Scheme 5 | Design cost | 5 | 6 | 5 | 6 | 5 | 5 | 6 | |
| | Production cost | 6 | 6 | 5 | 6 | 5 | 6 | 5 | |
| | Cost of ancillary facilities | 6 | 5 | 6 | 6 | 6 | 5 | 6 | |
| | Maintenance cost | 5 | 5 | 4 | 5 | 5 | 5 | 5 | 5.27 |
| | Scoring results | 5.46 | 5.40 | 4.72 | 5.56 | 5.16 | 5.30 | 5.26 | |

**Table A3.** Evaluation score of the comprehensive indicator "harms".

| Design Scheme | Project Indicators | Reviewer 1 | Reviewer 2 | Reviewer 3 | Reviewer 4 | Reviewer 5 | Reviewer 6 | Reviewer 7 | The Final Score |
|---|---|---|---|---|---|---|---|---|---|
| Scheme 1 | Reducing the degree of harmful function | 7 | 7 | 6 | 7 | 8 | 7 | 6 | 5.46 |
| | Reducing the number of harmful functions | 6 | 7 | 6 | 6 | 7 | 6 | 7 | |
| | Existing pollution | 5 | 6 | 5 | 6 | 5 | 5 | 6 | |
| | Existing risk | 4 | 5 | 4 | 4 | 5 | 4 | 5 | |
| | Scoring results | 5.11 | 5.97 | 4.97 | 5.64 | 5.58 | 5.11 | 5.83 | |
| Scheme 2 | Reducing the degree of harmful function | 4 | 5 | 4 | 4 | 5 | 5 | 4 | 6.37 |
| | Reducing the number of harmful functions | 5 | 6 | 5 | 5 | 4 | 6 | 5 | |
| | Existing pollution | 8 | 7 | 7 | 8 | 8 | 7 | 7 | |
| | Existing risk | 6 | 5 | 6 | 6 | 6 | 5 | 5 | |
| | Scoring results | 6.70 | 6.14 | 6.17 | 6.70 | 6.76 | 6.11 | 5.92 | |
| Scheme 3 | Reducing the degree of harmful function | 5 | 4 | 4 | 5 | 4 | 5 | 4 | 5.28 |
| | Reducing the number of harmful functions | 5 | 5 | 6 | 5 | 6 | 5 | 5 | |
| | Existing pollution | 5 | 6 | 5 | 5 | 6 | 6 | 5 | |
| | Existing risk | 6 | 6 | 5 | 6 | 5 | 5 | 6 | |
| | Scoring results | 5.25 | 5.64 | 4.94 | 5.00 | 5.47 | 5.53 | 5.11 | |
| Scheme 4 | Reducing the degree of harmful function | 5 | 5 | 4 | 5 | 4 | 5 | 5 | 5.86 |
| | Reducing the number of harmful functions | 5 | 4 | 5 | 5 | 6 | 5 | 5 | |
| | Existing pollution | 7 | 6 | 7 | 7 | 6 | 6 | 7 | |
| | Existing risk | 6 | 5 | 5 | 5 | 6 | 5 | 5 | |
| | Scoring results | 6.31 | 5.45 | 5.92 | 6.06 | 5.72 | 5.53 | 6.06 | |
| Scheme 5 | Reducing the degree of harmful function | 4 | 5 | 5 | 4 | 5 | 6 | 5 | 5.61 |
| | Reducing the number of harmful functions | 5 | 5 | 6 | 5 | 5 | 5 | 6 | |
| | Existing pollution | 6 | 6 | 5 | 7 | 6 | 5 | 6 | |
| | Existing risk | 5 | 6 | 6 | 5 | 6 | 6 | 5 | |
| | Scoring results | 5.39 | 5.78 | 5.53 | 5.96 | 5.78 | 5.39 | 5.61 | |

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
