# Peer review of "A Trimming Design Method Based on Bio-Inspired Design for System Innovation"

_applsci, doi:10.3390/app11094060_

Round 1

Reviewer 1 Report

The paper presents an attempt to provide a systematic support in solving design problems by biologically inspired ideas. There is definitely something interesting in the attempt to make finding the biological analogy simple by function model of the phenomenon. In general, I have never seen using function modelling for biological systems and if the authors are the first to introduce it, it is definitely smth new. The case study in interesting because every design case is interesting.   

However, the presented method seems to be based on much subjective evaluations, like quantified ranking, finding the analogy, choosing the best idea etc. It seems that each time the method is applied to another system or, more important, by another engineers there will be no guarantee that it will work. In short, there is an evidence that the new design methods is suggested, but no scientific evidence that this method of design can be applied by anybody (not by the authors) and, more important, the statistics of this application shows that it reliably generates really good results. To do so I would recommend the authors either to collect the results of application of the method to many design problems, or, ideally, to teach other engineers this method and collect their opinions with statistically representative amount. Another way of rewriting the paper could be the present not the new method, but a Design Case Study, to make the design case the center of the paper and, in the story of design, explain how did you get the ideas by TRIZ, BID, trimming and so on.

10. Trimming does not seem to be as know as you do not need explain what it is. If it were specific TRIZ-community journal, yes, but for general public it is not clear from the first strings of the paper what is the object of discussion

34. (and further) I would argue that TRIZ is invention method, not innovation method. Calling TRIZ innovation method is like calling a toothbrush as "body cleaning tool"

41. I am not sure, but I think the sentence requires passive mode "Trimming is applied..."

76. What is CECA? First met, it should have been decoded.

98. Literature review. General comments: it is not clear what was the criteria to select the papers to build the state of art. How we can know, that all important works on trimming or BID have been reviewed? There are several errors/typos in the the literature list, for example (1)'s author is not Simon, L. but Litvin, S. (798) Bibliography should be presented in more accurate form. (841). Unfortunately, many of sources are either non peer-reviewed or from research conferences with relaxed scientific level, like TRIZfest or TRIZcon,

188. "of function-oriented modeling methods" not clear what method is meant. In TRIZ there are FM (function modelling) and FOS (function oriented search). But FO is not known.

234. "its effectiveness closely depends on designers’ own knowledge limitations and experiences". This statement does not seem to have any support from the reviewed literature. I am afraid that effectiveness of trimming has not been discussed in literature in general.

243. "provides a function-orient design scenario" not clear what is "function-orient"

250. "trimming enhances the process of BID by putting forward clearly functional requirements for biological solutions" It is not clear how trimming can make BID more effective, because in BID the efficiency is based on the knowleadge of this or that biological phenomena. 

263 (Fig) In general "Target component" in TRIZ is the object of the main function of the system, this notation is confusing.

277. It is necessary to say that in general in Trimming this way of assigning of weights is absolutely subjective, there is no scientifically proven (nor statistically proven) evidence that this quantitative ranking of components can actually help to do more successful trimming. it makes the whole "Step 3." (293+) just a speculation. Why it is "-5" and "half of it"? Why not -23 and -8?

291. (Fig) CECA is presented in confusing from, it contains a combination of statements ("Water turn bad") and processes ("Heat water"). It is also unusual to see components in the same CECA ("Water"), it is not clear what kind of cause-effect can be establised btw, for example, "Component" and "Statement". In classical CECA there are only facts of statements. For example "Water is overheated" can result in "High pressure in tank" and it can result in "Explosion" 

301. Formula 1 is not found.

359. "main functions" Main function has specific meaning in Function Modelling, it is better not to use this term.

463. What is AHP? 

602. Some function in the Fig should be revised. "Connect" or "Unbalance" do not seem to be legitimate according to the definition of the function.

743. "three experts in the mechanical field are invited to score" it not really a reliable and reproductive way to detect the best solution. 

Author Response

Response to Reviewer 1 Comments

The paper presents an attempt to provide a systematic support in solving design problems by biologically inspired ideas. There is definitely something interesting in the attempt to make finding the biological analogy simple by function model of the phenomenon. In general, I have never seen using function modelling for biological systems and if the authors are the first to introduce it, it is definitely smth new. The case study in interesting because every design case is interesting.

Point 1: However, the presented method seems to be based on much subjective evaluations, like quantified ranking, finding the analogy, choosing the best idea etc. It seems that each time the method is applied to another system or, more important, by another engineers there will be no guarantee that it will work. In short, there is an evidence that the new design methods is suggested, but no scientific evidence that this method of design can be applied by anybody (not by the authors) and, more important, the statistics of this application shows that it reliably generates really good results. To do so I would recommend the authors either to collect the results of application of the method to many design problems, or, ideally, to teach other engineers this method and collect their opinions with statistically representative amount. Another way of rewriting the paper could be the present not the new method, but a Design Case Study, to make the design case the center of the paper and, in the story of design, explain how did you get the ideas by TRIZ, BID, trimming and so on.

Response 1: This paper proposes a design method that can be used for system design, and proves this point in the case. However, no innovative design method is unchanged, because the process of innovation is a complex process, which requires subjective participation of people. Therefore, different people using the same method will have different results, which is objectively difficult to change. We agree with your suggestion very much. In the following study, we expect to make the proposed method more versatile by refining some details of the method proposed in this paper, and prove this result through more cases. However, due to the limitation of space, we can only select a representative case in this paper to verify the feasibility of the proposed design method.

Point 2:10.Trimming does not seem to be as know as you do not need explain what it is. If it were specific TRIZ-community journal, yes, but for general public it is not clear from the first strings of the paper what is the object of discussion.

Response 2: Thank you very much for your advice, it is true that ordinary people may not understand what trimming is, I have made corrections in the article and added an explanation for trimming. I have changed “The selection of design knowledge determines the innovation of the technical scheme obtained by application trimming” into “The selection of design knowledge determines the innovation of the technical scheme obtained by application trimming(A tool for problem analysis and problem solving in TRIZ)”. -------(line 10, in blue)

Point 3: 34.(and further) I would argue that TRIZ is invention method, not innovation method. Calling TRIZ innovation method is like calling a toothbrush as "body cleaning tool"

Response 3: Maybe it is my unclear expression that makes you have some deviation in understanding. My original article is “As a method designed for solving refined design problems, trimming originates from the Theory of Inventive Problem Solving (TRIZ) and becomes a widespread method applied in product innovations.”, What I mean in the article is that trimming is an innovative method, and it originated from TRIZ. This is not to say that TRIZ is an innovative method. If you still find there is something wrong with the expression here after reading it again, please leave me your suggestions for revision, and I will definitely consider it carefully again.

Point 4: 41. I am not sure, but I think the sentence requires passive mode "Trimming is applied..."

Response 4: This one does have a problem with misrepresentation and I have changed "Trimming applies " to "Trimming is applied" in the article -------------(line 45, in blue)

Point 5:  76. What is CECA? First met, it should have been decoded.

Response 5: It has been explained in the article that "CECA" is changed to "CECA(Cause Effect Chain Analysis)" --------------------------------------------------------(Line 80 , in blue)

Point 6:  98.Literature review. General comments: it is not clear what was the criteria to select the papers to build the state of art. How we can know, that all important works on trimming or BID have been reviewed? There are several errors/typos in the the literature list, for example (1)'s author is not Simon, L. but Litvin, S. (798) Bibliography should be presented in more accurate form. (841). Unfortunately, many of sources are either non peer-reviewed or from research conferences with relaxed scientific level, like TRIZfest or TRIZcon.

Response 6: Thank you very much for your careful reading.

 First of all, according to your question, I searched the literature of BID or trimming again, and added the relevant literature retrieved into the article.(line 241-261, in blue)

 Secondly, I have corrected the errors related to the format of the references.----------------------------- (Please refer to the references in the article for specific modifications, line 918-1021, in blue)

Point 7: 188. "of function-oriented modeling methods" not clear what method is meant. In TRIZ there are FM (function modelling) and FOS (function oriented search). But FO is not known.

Response 7:  I didn't make it clear. There really is no such thing as "function-oriented modeling" in TRIZ, so I have corrected the error in the article by changing "function-oriented modeling methods" to "function modeling method".----------------(line 181, line195, in blue)

Point 8: 234. "its effectiveness closely depends on designers’ own knowledge limitations and experiences". This statement does not seem to have any support from the reviewed literature. I am afraid that effectiveness of trimming has not been discussed in literature in general.

Response 8:  Thank you very much for your careful reading. There are indeed some problems in the expression here. The limitation of the effectiveness of trimming by the designer ’ s knowledge and experience is the conclusion I draw after reading some literature. I have added the references I have read for reference. To avoid ambiguity, I have changed “closely depends ” to “ partly depends ”  --------------------------------------( Line 265, in blue )

Point 9:  243."provides a function-orient design scenario" not clear what is "function-orient"

Response 9: . Here I am trying to express : this paper combines trimming with BID, takes the missing function of the system after trimming as the trimming point, and uses it as the keywords to find biological knowledge, and constructs the functional model of biological prototype. On this basis, the functional model of the new system is constructed to obtain a new design scheme.  Trimming provides an application scenario for BID, which is function-based, so I wrote “ this design method provides a function-oriented design scenario for the application of biological knowledge in clipping problem solving ”.

Point 10: 250."trimming enhances the process of BID by putting forward clearly functional requirements for biological solutions" It is not clear how trimming can make BID more effective, because in BID the efficiency is based on the knowleadge of this or that biological phenomena. 

Response 10:  This may be the problem of my inaccurate expression. I want to express that trimming creates the use scene for BID. The application of BID in the trimming process enhances the application of BID in the design process. Therefore, I have changed “ trimming enhances the process of BID  by putting forward clearly functional requirements for biological solutions. ” to “ trimming improves the application of BID in solving creative problems by putting forward clearly functional requirements for biological solutions. ”----------------------------------------------------------------------------------------------------(line 281, in blue)

Point 11:  263 (Fig) In general "Target component" in TRIZ is the object of the main function of the system, this notation is confusing.

Response 11: Thank you very much for your reading. Here I want to express that after the trimming priority is determined, the components with low function value should be selected for trimming, so I call the components that need to be trimmed as "target components". If you think there is a problem with such an expression, then I will change the expression. I have changed "Target Component" to "Trimming Component".  ----------------------------------------------------------------------------(Modified in Figure 1, line 385, 386, 387, 388, 733,734, in blue )

Figure 1. Methodology of the trimming based biological inspired innovative design.

Point 12: 277. It is necessary to say that in general in Trimming this way of assigning of weights is absolutely subjective, there is no scientifically proven (nor statistically proven) evidence that this quantitative ranking of components can actually help to do more successful trimming. it makes the whole "Step 3." (293+) just a speculation. Why it is "-5" and "half of it"? Why not -23 and -8?

Response 12:

  • Indeed, as you said, there is some subjectivity in the way of weight allocation. To avoid this subjectivity, I quoted the score allocation method in Literature [13], which calculated the function level value of each function by assigning values to functions. In order to be more clear, I have changed the original article to " In order to calculate the functional values of each component, this paper uses the method in reference13to allocate the functional level scores of components.  ". -------------------------------------------------(line 310、311,in blue)

  • As for the reason of "-5" and "-2.5", the score of one harmful function is "-5". As shown in Figure 2, the heater and water tank jointly act on the harmful effect of "increasing the cost of energy/material" through the connection mode of "and". Therefore, the function values of these two components under this harmful function are "-2.5" and "-2.5" respectively. In order to be more clear, I have revised it in the original article.

Here is the modified sentence: " in the illustrative case shown in Figure 2, the heater and water tank components work together through the " and " relation for harmful function of " increase the costs of energy / material " with a score of " -5 ", so the functional values of the heater and water tank are " -2.5 ". "-----------------------------------------(line 334-337, in blue)

Point 13:  291. (Fig) CECA is presented in confusing from, it contains a combination of statements ("Water turn bad") and processes ("Heat water"). It is also unusual to see components in the same CECA ("Water"), it is not clear what kind of cause-effect can be establised btw, for example, "Component" and "Statement". In classical CECA there are only facts of statements. For example "Water is overheated" can result in "High pressure in tank" and it can result in "Explosion" 

Response 13:

As you said, Figure 2 is not a classical CECA diagram. This paper aims to establish the causal relationship between components and functions in the system on the basis of the analysis results of CECA, so as to construct the system function tree based on CECA, aiming to calculate the functional values of each component in the system.

   In order to facilitate readers to understand the CECA system function tree, this paper takes the water heater as an example to illustrate. First of all, the analysis is carried out according to the way of CECA, with the "problems in water tank" as the starting point to find out what causes the problems of the water tank, until the root cause of the problems of the water tank is found. In this case, it is believed that high energy/material consumption, poor water quality, insufficient heating, water leakage and so on are the reasons for the problems of the water tank. These four causes are described in terms of functions in Figure 2, and a cause-and-effect diagram is established between components and functions. Maybe I did not express it clearly enough. Now I have corrected the functions in the box in Figure 2 by using "verb + noun" again. and the title of Figure 2 was changed from " Cause Effect Chain Analysis of design problems in water heater product." to " Function tree of water heater product design problem based on CECA ".  If you still feel that there is a problem here, I will revise it carefully again.-----------------------------------------------------------------------(line 327, in blue)

Figure 2.  Function tree of water heater product design problem based on CECA

Point 14:   301. Formula 1 is not found.

Response 14: The previous formula (1) is now put in front of the following paragraph-------------------------------------------------------------------------------------------------------------(line 338)

Point 15:  359. "main functions" Main function has specific meaning in Function Modelling, it is better not to use this term.

Response 15: In the process of determining keywords in this paper, the first thing to do is to determine the useful functions missing from the system after trimming, and then to find the relatively major functions in the missing functions and abstract them as keywords. Therefore, I use "main function" to express them, It's really problematic to use “main function”. I have changed "main function" to "target function".--------------(line 405,407,408,413,744, in blue)

Point 16:  463. What is AHP? 

Response 16: AHP has been explained in The article, changed to "AHP (Analytic Hierarchy Process)", and references have been added.-------------(line 514, in blue)

Point 17: 602. Some function in the Fig should be revised. "Connect" or "Unbalance" do not seem to be legitimate according to the definition of the function.

Response 17: Thank you very much for your suggestions. There are indeed some problems in the description of functions in the figure, so I have made some modifications to the functions in the figure, such as changing "connect" to "fasten" and changing "unbalance" to "shake".   

Figure 9. Functional model of the original type steel tape armoring machine.

Point 18: 743. "three experts in the mechanical field are invited to score" it not really a reliable and reproductive way to detect the best solution. 

Response 18: Inviting three experts in the mechanical field to rate the design is not a reliable way to evaluate. So I have  changed it in the article, and through invite relevant experts, engineering and technical personnel, maintenance personnel and a line of technical personnel and so on various aspects of relevant personnel evaluation team, using the analytic hierarchy process(AHP) and expert evaluation method evaluation method to evaluate the design scheme, through the final evaluation result to judge whether the system ideal of the new design was improved.------------(I have corrected step C5 in the theory part and step C5 in the case study part, line 585-661, line 857-869, in blue )

Others:

 (1) Since new references are added, the order of references in the paper has also been adjusted

(2) In the process of rearranging, I also found some other mistakes and corrected them in the whole article. ------------------------(The modification process has been marked, in blue)

Thank you for reading and suggesting this paper in your busy schedule. If you still feel there are problems, please leave your valuable comments. I will revise it carefully again!

Reviewer 2 Report

The work shows the use of BID in applying Trimming: A TRIZ principle. The study presents a case study demonstrating the application of their work. The method shown by the authors is interesting. My section-wise recommendations for the work are as follows:

Introduction:

  • I found the introduction difficult to read for someone who is not knowledgeable in the field. Certain terms, like ‘trimming’ itself, may need defining.
  • Some clauses may require more evidence to ensure clarity For eg., line 70, says ‘performance problems’ can be transformed to ‘functionality problems’. An example or a reference to previous work seems to be required here.

Literature Review:

  • Can the authors also provide examples of trimming observed in biology? It would further make the introduction/ literature review more interesting.
  • I find a lot of similarity between the TRIZ principle of trimming and the function-based principle of function-sharing (simultaneous implementation of multiple functions using a single component). I would encourage the authors to link the existing function-sharing works, especially from BID, to this work. There are some works on function-sharing BID that use functional-models. You work may benefit from showing how your work is different from those works in BID.

Approach:

  • Line 273: I’m having a hard time understanding what exactly is ‘ideality’. Also, line 278: who decides the required ideality for a system?
  • You might want to add more details to how your approach is carried out. For example, (line 277) you specify the four categories of functions and then assign a number to each category. It’s unclear what the numbers are for. How are they decided, who decides them?
  • Figure 2: you specify functional-basis in your literature review, but seems you haven’t used it. You can try to stick to the ‘verb-noun’ notation for standardization.
  • Line 298: You might want to show weighted values on Figure 2.
  • Line 313-315 are hard to follow.
  • Line 338: I didn’t understand what you meant by ‘more than one component needs to be deleted’ ,Are you implying that trimming is required to be done on multiple components?
  • Line 338: Is the fuzzy set already been used in an existing study? If yes, please cite that. If no, presenting such usage is another contribution of this work.
  • 419: The numbers scales selected for quantifying different variables seem mostly subjective.
  • Here again, I see a lot of similarity between trimming and function-sharing. Works that represent the causal relationships between functions, behaviors and structures to implement bio-inspired function-sharing exist in the literature. It might be worth to relate this TRIZ methodology with a function-based methodology to validate the results.

Case Study:

  • The case study is good. But you might want to check with the figure numbers. Figure 9 doesn’t seem like a functional model.
  • Figure 10: It might be useful to have legend on each figure to improve readability.
  • Figure 11: Please label the subplots. (a) and (b) are barely readable.
  • Section c5 might need more information. For example, you may require to do an inter-rater agreement analysis for data in Table 5. Also, the evaluation index ‘negative impact’ seems ambiguous.

Conclusions:

  • I find the conclusion to be a little unclear. The authors might want to clearly write their important findings.
  • Line 771: Function-modeling make not be always graphic but can be coded up.
  • Rather than providing general limitations in BID and trimming research, please reflect of the specific limitations of your approach.

Other:

  • Please ensure that the full-forms of the abbreviations are specified upon the first introduction of the abbreviation. For eg. CECA (Line 76), MBE(Line 225).
  • There are a few typos here and there, and can be identified by proof-reading. For eg. Line 202, 266, 365
  • Line 594: I guess you mislabeled the functional model as figure 8. Also, I don’t see a functional model in the manuscript.

Author Response

Point 1:   Introduction:

  • I found the introduction difficult to read for someone who is not knowledgeable in the field. Certain terms, like ‘trimming’ itself, may need defining.

Response 1: Thank you very much for your advice. The trimming is not explained clearly in the introduction part, so the definition of trimming is added in the paper: “ Trimming is to ensure the realization of system functions by removing some components of the system and redistributing useful functions with residual system or super-system resources1. ”--------------------------------------------------------------------------------------------------------(line 35-37, in blue)

Point 2:  ·Some clauses may require more evidence to ensure clarity For eg., line 70, says ‘performance problems’ can be transformed to ‘functionality problems’. An example or a reference to previous work seems to be required here.

Response 2:  There may be some ambiguity here, but what I want to say is: The process of trimming is firstly to analyze the problems of the system and remove some components that may cause problems in the system. However, the trimming process may not only remove the components, but also lead to the absence of some useful functions in the system, transforming the original performance problems into functional problems. The next step after trimming is to use system or super-system resources to reallocate missing useful functions. That's why I said tailoring can turn a performance problem into a functional one. In order to make this kind of statement more valid, I have added relevant literature in the article to support it.-------------(line 75, in blue)

Point 3:  Literature Review:

  • Can the authors also provide examples of trimming observed in biology? It would further make the introduction/ literature review more interesting.

Response 3:  Thank you very much for your reading. I have searched the literature again, and so far I have not found any other relevant studies that combine trimming and BID. I am sorry that I have failed to cite other examples of trimming applied in biology.

Point 4: I find a lot of similarity between the TRIZ principle of trimming and the function-based principle of function-sharing (simultaneous implementation of multiple functions using a single component). I would encourage the authors to link the existing function-sharing works, especially from BID, to this work. There are some works on function-sharing BID that use functional-models. You work may benefit from showing how your work is different from those works in BID.

Response 4:  I have read relevant literature on function sharing, and it is true that biologically inspired function sharing has some similarities with the trimming method based on function sharing mentioned in this paper. I have added and summarized related studies on feature sharing in the literature review.---------------------------------------(line 241-261,in blue)

Point 5:  Approach:

  • Line 273: I’m having a hard time understanding what exactly is ‘ideality’. Also, line 278: who decides the required ideality for a system?

Response 5:  Ideality refers to the ratio of benefits to (costs+ Harms), expressed in a formula, which is described in the article. (line 109-111, line 594-598).

The formula states that the ideal level of a product or system is directly proportional to the sum of its benefits and inversely proportional to all its costs and all its harms. Raising the numerator, lowering the denominator, or increasing the ratio are all ways to raise the ideal level. The ideality of the system is determined by benefits, costs and harms. Maybe my expression is not clear enough, I have changed "which defined as perceived Benefits/ (Cost + Harm) to play the indicator of trimming16" to "and the ratio of perceived benefits / ( costs + harms ) is usually used as an indicator to judge whether the system ideality is improved17 ”----------------------------------------------------------------------------------------(line 109-111, in  blue)

Point 6:  You might want to add more details to how your approach is carried out. For example, (line 277) you specify the four categories of functions and then assign a number to each category. It’s unclear what the numbers are for. How are they decided, who decides them?

Response 6: The purpose of assigning scores for harmful, useful, insufficient and excessive functions in this paper is to calculate the functional values of each element in the system. The method of assigning scores is to directly quote the method in Literature [13], which can be of different values.  In Literature [13], the method of assigning scores is to define useful functions as “5” and harmful functions as “-5”.The insufficient function is “3”, and the excessive function is “-3”.

In order to avoid the subjectivity of the score allocation, I directly quoted the literature [13] in the article. Maybe I did not express it clearly, now I have reedited this sentence and changed it to “ In order to calculate the functional values of each component, this paper uses the method in reference13 to allocate the functional level scores of components. ”--------------------------------------------------------------------------------------------------(line 310-311,  in blue)

Point 7: Figure 2: you specify functional-basis in your literature review, but seems you haven’t used it. You can try to stick to the ‘verb-noun’ notation for standardization.

Response 7:  Thank you very much for your suggestions. The function description in Figure 2 is indeed confused. According to your opinions, the function has been standardized in the form of "verb + noun".---------------------------- (the details have been modified in Figure 2).

Figure 2

Point 8:  Line 298: You might want to show weighted values on Figure 2.

Response 8:  The function value of the component has been added next to the component according to your suggestion --------------------------------(as shown in figure 2, line 325)

Point 9:·Line 313-315 are hard to follow.  

Response 9: Thank you very much for your reading. There is a problem of unclear expression here. First of all, at the beginning, we assign values to the four functions of harmful, excessive, useful and insufficient. Among them, harmful: -5, excess: -3, useful: 5 and insufficient: 3. Based on the CECA system function tree, The harmful effects and excessive functions located at the root of the functional tree have a higher functional value, because they are the fundamental functions that cause system problems. Since the harmful and excessive function level scores are negative numbers, the harmful and excessive functions at the root are considered to be higher than the harmful and excessive functions at the top of the function tree. Among the useful parts containing useful functions and insufficient functions, the reason why the functions at the top level have higher priority is because in the function tree of the useful parts, the top functions are directly related to the functions required by the system. The function level scores of functions and insufficient functions are positive. The useful functions at the top level have lower function values, and the trimming is done from the component with the lowest function value. Therefore, it is expressed here that top-level nodes have higher priority when calculating their function values.

Now I have re-edited this paragraph and changed it to " In the CECA-based system function tree, harmful and excessive functions located at the root of the function tree have higher function values than those located at the top of the function tree, because they are the foundations of the whole causal chain for causing the design problems. For the useful section containing useful and insufficient functions, because the top-level function is directly related to the functions required by the system, the top-level useful function has a lower function value.” ----------------------------------------------------------------------------(line 346-352, in blue)

Point 10:  Line 338: I didn’t understand what you meant by ‘more than one component needs to be deleted’ ,Are you implying that trimming is required to be done on multiple components?

Response 10:  As you understand, this process may indeed trim multiple components. The trimming process starts with the component with the lowest function value. If the component with the lowest function value is trimmed, but the component that caused the system problem still exists. We need to continue trimming until all components that cause system problems have been trimmed. Therefore, the expression in the article is "more than one component needs to be deleted in the actual system ". In order to make the expression more clear, I added "may" to this sentence and changed this sentence to " more than one component may need to be deleted in the actual system ". --------------------------------------------------(line 384, in blue)

Point 11:  Line 338: Is the fuzzy set already been used in an existing study? If yes, please cite that. If no, presenting such usage is another contribution of this work.

Response 11: Thank you very much for your suggestion. There are already literatures that use fuzzy sets in the similarity calculation. Here I am missing the citations of related literatures, so I have added the citations of two references. --------------(line 436, in blue)

Point 12:  419: The numbers scales selected for quantifying different variables seem mostly subjective.

Response 12: The method of assigning weights to the three indicators of compatibility、completeness and feasibility in the article is indeed subjective, and the expression is not clear enough. According to the relevant analysis of the three indicators of compatibility, completeness and feasibility in the article, it can be known that compatibility and feasibility are  two more important indicators. Here we invited 5 scholars  to assign weights to the three indicators based on the above analysis, and finally get the average score, and the weights for compatibility, completeness, and feasibility are 0.4, 0.2, 0.4.-------------------------------------------------------------------------------------------------------------------------(line 467-471,in blue)

Point 13:  Here again, I see a lot of similarity between trimming and function-sharing. Works that represent the causal relationships between functions, behaviors and structures to implement bio-inspired function-sharing exist in the literature. It might be worth to relate this TRIZ methodology with a function-based methodology to validate the results.

Response 13:  Thank you very much for your reading and suggestions. The literature related to function sharing has been put in the article and explained accordingly. --------------------------------------------------------------------------------------------------------------(line 241-261,in blue)

Point 14: Case Study:

  • The case study is good. But you might want to check with the figure numbers. Figure 9 doesn’t seem like a functional model.

Response 14:  This is indeed the problem of my number indicating wrong. The "9" in line 708 has been changed to "10".----------------(line 709, in blue )

Point 15:   Figure 10: It might be useful to have legend on each figure to improve readability.

Response 15: The lack of a legend in the picture does have a problem of poor readability. According to your suggestion, a legend description has been added to Figure 10, and the title of Figure 10 has been changed from ”CECA analysis result of original type steel tape armoring machine (partial)  ” to “ Function tree of original model steel tape armoring machine based on CECA (partial)  ” .---------------------------------------------------------------------------------------------------(The specific modification is shown in Figure 10, line 713, in blue)

Figure 10

Point 16:  Figure 11: Please label the subplots. (a) and (b) are barely readable.

Response 16: Figure 11 has been re-modified to enhance the readability of (a) and (b). ----------------------------------------------------------(The specific modification is shown in Figure 11)

Figure 11

Point 17:  Section c5 might need more information. For example, you may require to do an inter-rater agreement analysis for data in Table 5. Also, the evaluation index ‘negative impact’ seems ambiguous.

Response 17:  Thank you very much for your suggestions. I have added more information in step C5 and updated the relevant indicators and weight values in Table 4. Previously, there was a certain degree of subjectivity in the classification of evaluation indicators and the determination of weight values:

     First of all, the previous evaluation indcator is to analyze and further subdivide the three elements in the idealization level formula to obtain the final evaluation indicator, some of which are subjective judgments, and now, I have been modified it. With reference to the specific meaning of the idealized level formula and related research in other documents and summarized, the current evaluation indicator system is obtained, as shown in Table 4. ---------------------------------(The specific modification is shown in table 4, line 617, indicated in blue)

Secondly, part of the weight value distribution was subjectively judged before. Now referring to the research of others, the weight value is re-allocated using the analytic hierarchy process(AHP). The specific weight value determination process has been explained in the article. ---------------------------------------------------------------------(line593-661, line 858-870, in blue, I have modified both c5 in the theory part and c5 in the case part)

Point 18:  Conclusions:

  • I find the conclusion to be a little unclear. The authors might want to clearly write their important findings.

Response 18: Thank you very much for your careful reading, I have revised the conclusion part. ------------------(line 871-906, in blue)

Point 19:  Line 771: Function-modeling make not be always graphic but can be coded up.

Response 19:  The functional model is indeed not only in graphics, but also in coded form. I agree with this point very much. In the line 181-194 of the article, I also describe the functional model.

 But what I want to explain here is:  because the functional model of the biological prototype used in this article is shown in the form of graphics, in this conclusion, the first limitation mentioned is to establish the functional model of all biological prototypes. It is an arduous task, and in this work we can only select a part of biological prototypes, model them and obtain graphical functional models. That's why I used "graphic function model" to describe in the article. If you still feel that there is a problem here, please leave your valuable comments, and I will definitely revise it again.

Point 20: Rather than providing general limitations in BID and trimming research, please reflect of the specific limitations of your approach.

Response 20:  Thank you very much for your suggestion. In accordance with your suggestion, the limitations of this research have been further explained.-------------------(line 879-895, in blue)

Point 21: Other:

  • Please ensure that the full-forms of the abbreviations are specified upon the first introduction of the abbreviation. For eg. CECA (Line 76), MBE(Line 225).

Response 21: The CECA and MBE in the article have been modified, CECA (Cause Effect Chain Analysis), MBE (multi-biological effects). ----------------------(line 80, 233,  in  blue)

Point 22:   There are a few typos here and there, and can be identified by proof-reading. For eg. Line 202, 266, 365

Response 22: Thank you for your reminding. I have corrected the relevant mistakes in the article:

SAPPhIRE (Line 202): The word is taken from other sources, it's an abbreviation for “ State-Action-Parts-Phenomenon-Input-oRgan-Effects ”. So I explained “SAPPhIRE” in the article, changed it to “SAPPhIRE (State-Action-Parts-Phenomenon-Input-oRgan-Effects)”.------------------------------------------------------------------------------------------------------(line 207, in blue)

Post-trimming(line 266): I've changed “post-trimming system” to “ trimmed system”. And it has been modified in the full article.---------------------------------------(line 294, 170,17,43, 59, 155,  in blue)

Line 365: I have changed some of the words in this sentence, for example,Change “thesaurus” to “dictionary”, “such as engineering to biology dictionary” is put in parentheses.--------------------------------------------------------------------(line 402-404, in blue)

Point 23: Line 594: I guess you mislabeled the functional model as figure 8. Also, I don’t see a functional model in the manuscript.

Response 23: It is true that I have incorrectly indicated the number. It should be figure 9, and I been corrected  it in the article. ------------(line 703, in blue)

    And the functional model is shown in Figure 9 and Figure 15.

Others:

  • In the process of rearranging, I also found some other mistakes and corrected them in the whole article. ------------------------(The modification process has been marked, in blue)

 (2) Since new references are added, the order of references in the paper has also been adjusted

Thank you for reading and suggesting this paper in your busy schedule. If you still feel there are problems, please leave your valuable comments. I will revise it carefully again!

Round 2

Reviewer 1 Report

Thank you for your new version of the paper.

There were 2 types of comments in my review.

One type was mostly about presenting your ideas and research. In these terms the new version is definitely better, many editing issues have been addressed and resolved.

Another type of my comments required more research to be done. Either collecting more proofs that the method is correct and useful or more applications of the methods to other systems or objects. These issues are partially addressed in the new version, too, e.g. the amount of reviewers is increased. I personally believe that more to be done to prove that the method is reproducible by others and beneficial comparing to other methods or control group. (I can not agree that in science application of the same method by different people is allowed to deliver different results, otherwise the method is not in the domain of science.) But I need to leave the decision (is it proven enough or not proven enough) to the journal's editor.

Author Response

Point:Another type of my comments required more research to be done. Either collecting more proofs that the method is correct and useful or more applications of the methods to other systems or objects. These issues are partially addressed in the new version, too, e.g. the amount of reviewers is increased. I personally believe that more to be done to prove that the method is reproducible by others and beneficial comparing to other methods or control group. (I can not agree that in science application of the same method by different people is allowed to deliver different results, otherwise the method is not in the domain of science.) But I need to leave the decision (is it proven enough or not proven enough) to the journal's editor.

Response:Thank you very much for your advice. It is true that there are some limitations in our paper. At present, only one case is used to prove the effectiveness of the proposed method. In the future research, we will definitely continue to refine this method and apply it to more systems or objects.

The following are the modifications made in the article according to your opinions:

First of all, more evidence was collected to prove the effectiveness of the design method proposed in this paper. In step C5, the process of program comparison is added. First, three programs of steel tape armoring machines designed with other design methods are obtained through retrieval. The steel tape armoring machine program designed using the design method in this paper will be compared with the original armoring machine program and Other design schemes, and the five schemes are evaluated by the scheme evaluation process mentioned in the article to prove that the trimming design method is beneficial compared with other design methods. -------------------------------------------------------------------------------------------(Line867-877, in blue)

  Secondly, the literature “ Impact of TRIZ Learning on Performance in Biologically Inspired Design” is added, this literature proves that the combination of TRIZ and BID can improve the innovation ability of designers and generate more novel design schemes by means of experimental comparison.---------------------------------------------------------------------------(Line 901-905, in blue)

Other:

(1) An author’s unit information was filled in incorrectly before, and now it has been corrected. ---------------------------------------------------------------------------------------------------(Line 8, in blue)

(2) A sentence was added to step C5 of the theory section. --------------------(Line 601-604, in blue)

(3) Small revisions were also made to the conclusion part.------------------------- (Line 929, in blue)

(4) four references have been added.------------------------------------------- (Line 1112-1118, in blue)

(5) Because there are more comparisons of other schemes, supplements are made in the appendix. --------------------------------------------------------------------------------(Line 978, in blue)

(6) The format of all the Tables in the paper has been modified.

(7) The format of the paper has been re-checked and modified.

(8) The order of appendix and references is reversed

Changes made in this article are shown in blue.
